



# Characterization of offline analysis of particulate matter with
# FIGAERO-CIMS
Jing Cai[1,2,#], Kaspar R. Daellenbach[1,2,3,#,*], Cheng Wu[4,5], Yan Zheng[6], Feixue Zheng[1], Wei Du[1,2], Sophie L. Haslett[4],
Qi Chen[6], Markku Kulmala[1,2], Claudia Mohr[4,*]
[1] Aerosol and Haze Laboratory, Beijing Advanced Innovation Center for Soft Matter Science and Engineering,
Beijing University of Chemical Technology, Beijing 100029, China
[2] Institute for Atmospheric and Earth System Research, Faculty of Science, University of Helsinki, Helsinki 00014,
Finland
[3] Laboratory of Atmospheric Chemistry, Paul Scherrer Institute, Villigen, Switzerland.
[4] Department of Environmental Science, Stockholm University, Stockholm, 11418, Sweden
[5] Department of Chemistry and Molecular Biology, Atmospheric Science, University of Gothenburg, Gothenburg,
SE-412 96, Sweden
[6] State Key Joint Laboratory of Environmental Simulation and Pollution Control, Beijing Innovation Center for
Engineering Science and Advanced Technology, College of Environmental Science and Engineering, Peking
University, Beijing, 100871, China
[#] These authors contributed equally to this work.
*Correspondence to:* kaspar.dallenbach@helsinki.fi and claudia.mohr@aces.su.se
**Abstract:** Measurements of the molecular composition of organic aerosol (OA) constituents improve our
understanding of sources, formation processes, and physicochemical properties of OA. One instrument providing
such data at a time resolution of minutes to hours is the Chemical Ionization time-of-flight Mass Spectrometer
with Filter Inlet for Gases and AEROsols (FIGAERO-CIMS). The technique collects particles on a filter, which
are subsequently desorbed, and the evaporated molecules are ionized and analyzed in the mass spectrometer.
However, long-term measurements using this technique and/or field deployments at several sites simultaneously,
require substantial human and financial resources. The analysis of filter samples collected outside the instrument
(offline) may provide a more cost-efficient alternative and makes this technology available for the large number
of particle filter samples collected routinely at many different sites globally. Filter-based offline use of the
FIGAERO-CIMS limits this method albeit to particle-phase analyses, likely at reduced time resolution compared
to online deployments. Here we present the application and assessment of offline FIGAERO-CIMS, using Teflon
and Quartz fiber filter samples that were collected in autumn 2018 in urban Beijing. We demonstrate the feasibility
of the offline application with "sandwich" sample preparation for the identified over 900 organic compounds with
(1) high signal-to-noise ratios, (2) high repeatability, and (3) linear signal response to the filter loadings.
Comparable overall signals were observed between the Quartz fiber and Teflon filters for 12-h and 24-h samples,
but with larger signals for semi-volatile compounds for the Quartz fiber filters, likely due to adsorption artifacts.
We also compare desorption profile (thermogram) shapes for the two filter materials. Thermograms are used to
derive volatility qualitatively based on the desorption temperature at which the maximum signal intensity of a
compound is observed ($T_{max}$). While we find that $T_{max}$ can be determined with high repeatability for one filter type,
we observe considerable differences in $T_{max}$ between the Quartz and Teflon filters, warranting further investigation
into the thermal desorption characteristics of different filter types. Overall, this study provides a basis for
expanding OA molecular characterization by FIGAERO-CIMS to situations where and when deployment of the
instrument itself is not possible.
**1. Introduction**





Molecular information on organic aerosol (OA) composition is important for understanding the role that OA
plays in the atmosphere regarding its impacts on air quality, human health, and the climate (Daellenbach et al.,
2020; Huang et al., 2014; Cappa et al., 2012; Yao et al., 2018; Riipinen et al., 2012). Such data can be obtained
from offline filter collection and analysis in the laboratory using optical (e.g. Fourier transform infrared
spectroscopy, FTIR) and magnetic (e.g. Nuclear magnetic resonance spectroscopy, NMR) spectroscopy or, more
commonly, high-resolution mass spectrometer methods, which include gas/liquid chromatography coupled to
mass spectrometry (GC/LC-MS), ultrahigh-performance liquid chromatography coupled to Orbitrap mass
spectrometry and electrospray ionization mass spectrometry (ESI-MS) (Noziere et al., 2015). In contrast, online
mass spectrometers provide direct and in-situ information on particles' molecular composition, e.g. the filter inlet
for gases and aerosols coupled to a high-resolution time-of-flight chemical ionization mass spectrometer
(FIGAERO-HR-ToF-CIMS, Aerodyne Research Inc., US, hereafter FIGAERO-CIMS (Lopez-Hilfiker et al.,
2014)) or the extractive electrospray ionization time-of-flight mass spectrometer (EESI-MS) (Lopez-Hilfiker et
al., 2019). Since the particle-phase measurement by FIGAERO-CIMS is filter-based, it has the potential to be used
for offline analysis. Briefly, in the FIGAERO, particles are collected on a Teflon® (hereafter Teflon) filter and
analyzed via thermal desorption. When coupled to a high-resolution time-of-flight chemical-ionization mass
spectrometer (hereafter CIMS), molecular composition information of inorganic and organic aerosol compounds
that evaporate at temperatures up to 200 °C can be achieved. Having the advantage of combining molecular
composition and volatility information, the FIGAERO-CIMS has been widely used for measuring OA compounds
in many different environments including e.g. forests (Lopez-Hilfiker et al., 2016; Lee et al., 2016; Lee et al., 2018;
Mohr et al., 2019), rural and urban areas (Le Breton et al., 2019; Huang et al., 2019b; Cai et al., 2022), and indoor
air (Farmer et al., 2019).
Both online and offline techniques have their advantages and disadvantages and are associated with artefacts
(Turpin and Lim, 2001; Turpin et al., 2000). Online instruments generally allow for measurements at higher time
resolution, which is an advantage when studying rapid atmospheric processes, and no sample storage is needed
before analysis. However, the deployment of the FIGAERO-CIMS outside the laboratory requires a well-equipped
site that is easily accessible. In addition, long-term maintenance of these complex mass spectrometers needs
substantial human and financial resources. Therefore, deployments are often achieved only for short periods (i.e.
campaigns lasting from a couple of weeks to months), which limits the application of this technique for monitoring
and simultaneous measurements at multiple sites. Furthermore, FIGAERO gas-phase measurements have to be
interrupted regularly for particle-phase analysis in online usage, which could be a problem for measurements
requiring high time resolution data (e.g. chamber studies). Using the FIGAERO-CIMS for analyzing filters
collected elsewhere ("offline application") may therefore provide a valid alternative for long-term monitoring or
simultaneous measurements at multiple sites. Whereas the online FIGAERO-CIMS technique typically uses
Teflon filters to reduce interferences from the gas phase, Quartz fiber filters are widely used for offline sampling
of OA due to their high melting point and insolubility in water and typical organic solvents (Watson and Chow,
2002; Tao et al., 2017; Schauer et al., 2002; Gustafson and Dickhut, 1997). Up to now, only a few studies have
used the FIGAERO-CIMS in offline mode with Teflon filters (Siegel et al., 2020; Huang et al., 2019a), and an in-
depth characterization of the method is missing. The performance of Quartz fiber filters in FIGAERO-CIMS needs
to be assessed and compared to Teflon filters.
Here, we describe the application of FIGAERO-CIMS in offline mode for the analysis of particles deposited on
Teflon and Quartz fiber filters in urban Beijing during the autumn and winter of 2018. The filter deposition time
varies from 30 min to 24 h. We assess the performance of FIGAERO-CIMS for offline characterization of OA as
well as inorganic compounds and discuss background determination, reproducibility, and linearity of response for
the two filter types. We describe filter handling and offline analysis procedures and show the comparison of signals
from different mass loadings collected on both filter types. The utility of the FIGAERO for offline use is
demonstrated in this study. The potential to broaden its application for OA component measurements in future
research is also discussed.
**2.    Methods**



## 2.1 Filter sampling

The sampling site is situated on the west campus of the Beijing University of Chemical Technology (BUCT, 39°
56'31" N, 116°17'50" E). BUCT is located near the West Third Ring Road of Beijing, surrounded by residential
areas. A more detailed description of the sampling site can be found elsewhere (Cai et al., 2020; Kontkanen et al.,
2020; Liu et al., 2020; Yao et al., 2020; Fan et al., 2021; Guo et al., 2021). From November to December 2018,
samples of fine particulate matter with an aerodynamic diameter of up to 2.5 μm ($PM_{2.5}$) were collected by a four-
channel sampler (TH-16A, Tianhong Co., China) with a sampling flow rate of 16.7 L min$^{-1}$, installed on the rooftop
of a five-floor building (~20m above ground). Both Teflon (Zefluor® PTFE membrane, 1 μm pore size, 47 mm
diameter, Pall Corp., US) and Quartz fiber filters (7202, 47 mm diameter, Pall Corp., US) were collected
simultaneously at separate channels equipped with separate $PM_{2.5}$ cyclones of the sampler.
To investigate the influence of filter mass loadings and collection time on the signal response, the following filter
samples were taken: (1) 5 pairs of samples (Teflon/Quartz fiber filters) with 30 min deposition time on Dec 15,
2018 between 14:00 to 16:30 (Table 1). At the same time, an additional pair of Teflon/Quartz samples were
deposited for 2.5 hours using the other two separate channels of the sampler. (2) 12-h samples of Quartz/Teflon
filters from Oct 26 to Oct 30 and Nov 3 to Nov 24 (here only the Quartz filters from Nov 3 to Nov 16 were analyzed
(in total 27 pair of samples), shown in Table 1). (3) 24-h Quartz/Teflon samples from Oct 26 to Oct 30 and Nov 3
to Nov 25 (here only one pair of Teflon/Quartz filters was analyzed, shown in Table 1). During the last sampling
period, high $PM_{2.5}$ and relative humidity (RH) conditions prevailed (Nov 3:181 μg m$^{-3}$, 60%, and Nov 13: 227 μg
m$^{-3}$, 75%), and the channel of the 24-h sampling Teflon filter got clogged. Thus, only one pair of 24-h
Teflon/Quartz samples from this period was analyzed (Table 1).
Detailed information on the sampling protocol is listed in Table 1. Three pairs (Teflon/Quartz) of field blank
samples were also collected during the sampling period. Before sampling, Teflon filters were baked for 2 hours at
200 °C, which is much longer than the typical desorption time for FIGAERO-CIMS online usage (Ylisirniö et al.,
2021), and Quartz filters for 4.5 hours at 550 °C (Liu et al., 2016) in order to minimize contamination. After
sampling, samples were put in filter holders wrapped in pre-baked aluminum foils, individually sealed in a sealed
bag and stored in a freezer at -20 °C for 7 months until being analyzed in the laboratory.
To calculate the OA mass loadings of the samples, an online Time-of-Flight-Aerosol Chemical Speciation
Monitor (Aerodyne Research Inc., US, hereafter ToF-ACSM) equipped with a $PM_{2.5}$ lens and standard vaporizer
was operated during the sampling period at the same site. Details of the ToF-ACSM settings can be found in Cai
et al. (2022).

Table 1: Testing objectives, filter deposition dates and times, flows, filter material (T = Teflon, Q = Quartz fiber),
filter mass loadings of OA, number of samples, and number of sample repeats (filter punches) for the same filter.






| Testing objective | Sampling date | Sampling time | Filter material | OA loading [µg] per punch (punch diameter, area) | Number of samples/repeats |
|---|---|---|---|---|---|
| (1) Baseline subtraction tests, (2) reproducibility tests, (3) filter type comparison | Dec 15 14:00 – 16:30 (30 min-interval) | 30 min | T & Q | $1.7\times10^{-2}$ –$2.0\times10^{-2}$ (2 mm, 0.031 cm$^2$) | 1/1 |
| | Dec 15, 14:00 – 16:30 | 2.5 h | T & Q | $9.1\times10^{-2}$ (2 mm, 0.031 cm$^2$) | 1/3 for repeats |
| (1) Reheating tests, (2) filter type comparison | Nov 8 21:30– Nov 9 9:00 | 12 h | T & Q | $6.5\times10^{-1}$ (2 mm, 0.031 cm$^2$) | 1/1 |
| Reheating tests | Nov 12 21:30– Nov 13 9:00 | 12 h | Q | 0.75 (2 mm, 0.031 cm$^2$) | 1/1 |
| Reheating tests | Nov 13 21:30– Nov 14 9:00 | 12 h | Q | 1.2 (2 mm, 0.031 cm$^2$) | 1/1 |
| (1) Filter type comparison, (2) different ramping protocols for 2 mm punch, (3) linearity response for signals from different filter punch areas | Nov 24 9:30– 9:00 25 | 24 h | T & Q | 1.2 (2 mm, 0.031 cm$^2$) | 1/3 for repeats and 1/3 for different ramping protocols |
| | | | | 2.7 (3 mm, 0.071 cm$^2$) | 1/1 |
| | | | | 4.8 (4 mm, 0.13 cm$^2$) | 1/1 |
| | | | | 15 (7 mm, 0.38 cm$^2$) | 1/1 |
| Comparison of 12-h signals to ToF-ACSM | Nov 3 to Nov 16 | 12 h | Q | $5.0\times10^{-2}$ – 1.2 (2 mm, 0.031 cm$^2$) | 27/1 |




## 2.2 Offline application of FIGAERO-CIMS

### 2.2.1 Measurement approach

#### 2.2.1.1 FIGAERO-CIMS setup

The molecular composition of OA collected on the filter samples was characterized with FIGAERO-CIMS using iodide ($I^-$) as the reagent ion. In typical online FIGAERO-CIMS operation, particles are collected on a filter (Zefluor® Teflon filters) with a sampling time of a few minutes to hours and then thermally desorbed by a flow of temperature-controlled ultra-pure nitrogen (99.999 %) immediately following deposition. The thermally desorbed compounds are charged by clustering with $I^-$, which is typically generated through the exposure of methyl iodide to an X-ray or radioactive source for FIGAERO-CIMS ($Po^{210}$ in our study). In this study, we used the FIGAERO-CIMS in the laboratory to analyze filter samples collected earlier in the field. These samples were placed manually one by one in the dedicated filter holder of the FIGAERO-CIMS and the desorption procedure was started (see 2.2.1.3).

#### 2.2.1.2 Sample preparation and test design

Since the total particle mass collected on one filter was generally too large to be analyzed at once in its entirety by FIGAERO-CIMS (due to the risk of titration of the reagent ion), we only analyzed small circular punches of the collected filters. The default punching area was $3.1\times10^{-2}$ cm$^2$ (punch diameter $d$=2 mm). In addition, to test the linearity of response to sample mass loadings, punch areas for the same filter were varied between $3.1\times10^{-2}$ cm$^2$ ($d$=2 mm) and 0.38 cm$^2$ ($d$=7mm), resulting in variation in mass loadings by a factor of 10 (shown in Table 1). Since the filter punches were too small for the filter holder of the FIGAERO, we put them between two pre-baked originally sized ($d$=25 mm) Zefluor® Teflon filters ("sandwich technique", Fig. 1a). Field blanks were prepared analogously.

The OA mass loadings of the filter punches were estimated with the co-located ToF-ACSM in this study (details shown in Table 1). To test the performance of the method, we did the following tests (Fig. 1, Table 1): (1) reheating a few filters to determine backgrounds (see section 2.2.4), (2) assess different background subtraction methods, (3) reproducibility of signals from the same filter  (section 3.4), (3) the linearity of signal response from different punching areas from the same filter (section 3.4),  (3) comparing signals from different ramping protocols (section 2.2.1.3), (4) comparison between and offline FIGAERO-CIMS and online ToF-ACSM (section 3.5), (5) signals from different filter types (section 3.6), and (6) thermograms from different types of filters (section 3.7).

#### 2.2.1.3 Temperature ramping protocols

Reagent ion depletion is undesired as it can create non-linearities in the instrument response (Koss et al., 2018; Zheng et al., 2021). To avoid reagent ion depletion in FIGAERO-CIMS, the concentration of sample ions entering the instrument is controlled, typically by modifying the particle mass loading on the filter and/or the heating rate. While the particle mass loading can be varied easily when operating the FIGAERO-CIMS online through adjustment of sampling time and flow, in offline mode with pre-collected samples this can only be modified by the fraction of filter surface analyzed. For our Beijing filter samples, even when using the smallest punch sizes ($3.1\times10^{-2}$ cm$^2$), mass loadings of especially nitric acid ($HNO_3$) were still high enough to lead to titration of the reagent ion. We note that this can also be an issue for online measurements in presence of high nitrate concentrations, e.g. in highly polluted areas. In order to reduce reagent ion depletion between 60 °C to 105 °C desorption temperature, where $HNO_3$ exhibits a maximum signal, we used a heating protocol with a non-uniform temperature ramping procedure. Instead of ramping from room temperature to 200 °C with a constant heating rate, we divided the temperature ramp into several periods: (1) from room temperature (~25 °C) to 60 °C in 8 min





167 (4.4 °C min⁻¹), (2) from 60 °C to 105 °C in 15 min (3 °C min⁻¹), (3) from 105 °C to 200 °C in 12 min (7.9 °C min⁻¹). The ramp period was followed by a 20-minute soaking period (200 °C) to allow signals to go to background levels. We called this temperature ramping protocol non-uniform temperature ramping and used it as the default desorption procedure in this study. The maximum reagent ion depletion achieved in this way was ~35% for the samples with the highest mass loadings on a 2 mm punch, which was mostly used in this study. We also tested two alternative heating protocols:

> 1) Slow non-uniform temperature ramping: Same as the non-uniform ramping protocol, but with (2) slowed down to 1.5 °C min⁻¹. The total heating time for this protocol was 70 minutes, and the maximum reagent ion depletion was ~ 20%.
> 2) Uniform temperature ramping: The temperature was increased from room temperature to 200 °C in 31.5 min (5.7 °C min⁻¹). Including the 20 min soak, the total heating was 51.5 minutes, and the maximum reagent ion depletion was around 50%. In order to limit reagent ion depletion, the heating rate was 1.8–3.5 times slower than typical rates used for online FIGAERO-CIMS applications (10–20 °C min⁻¹ (Thornton et al., 2020)).

181 The 3 temperature ramping protocols are displayed in Fig. 1d. As different heating rates lead to different thermogram shapes and $T_{max}$ for individual compounds, we developed a correction method in an effort to be able to compare desorption-derived volatility for the different ramping protocols. This will be further discussed in section 3.3.

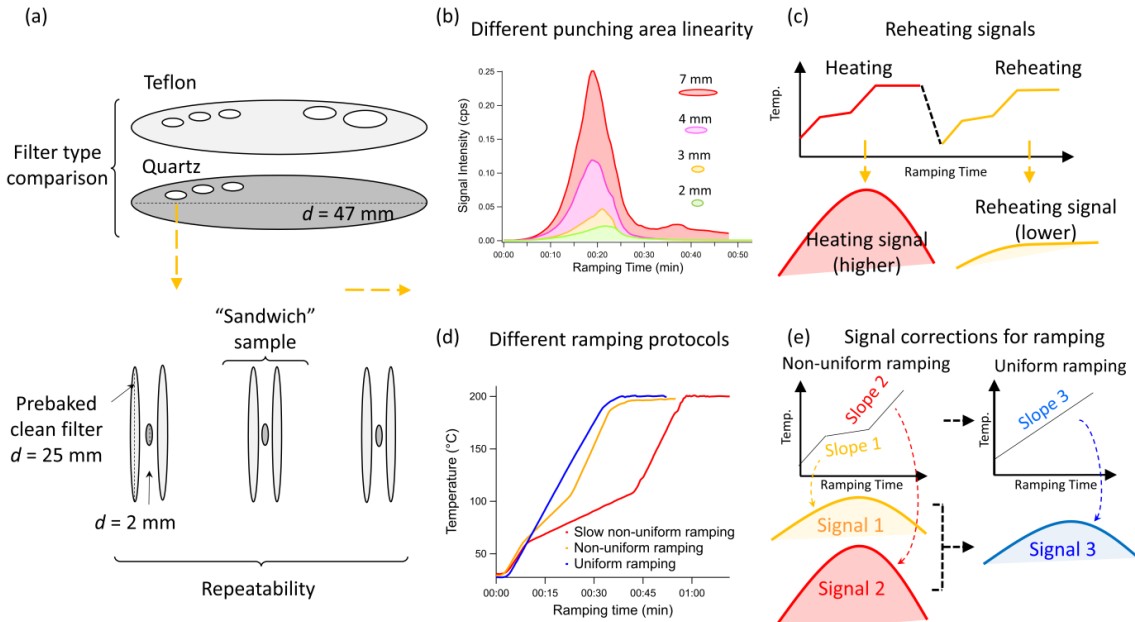

**Figure 1.** Schematic of the tests conducted in this study, (a) sample preparation using punching areas of different sizes of the Teflon and Quartz fiber filters and squeezing them between two original-sized filters for analysis, (b) signal intensities of different punching areas from the same sample with the same analytical procedure, (c) reheating tests by conducting two consecutive heating cycles, (d) different temperature procedures, and (e) signal intensity correction from non-uniform ramping to uniform ramping.





### 2.2.2 Data analysis

In this study, FIGAERO-CIMS data were analyzed with the Tofware package (v.3.1.0, Tofwerk, Switzerland, and Aerodyne, US) within the Igor Pro software (v.7.08, Wavemetrics, US). Mass accuracies of low- to high-mass species (~130 to 500 Da) were within ±10 ppm for all the samples. A total of ~1,200 peaks were found in the range of 46 and 500 Da, of which 916 were identified as organic species. Detailed information about the identified chemical compounds can be found in Cai et al. (2022). The total signal of a compound per filter sample, defined as the integrated signals ($Is$), was calculated by integrating the entire thermogram (ramping and soaking, normalized by the signals of $I^-$). Signals of the first 1.5 min of ramping and the last 1.5 min of soaking periods were excluded in order to remove potential interference from switching to and from the heating status. In this study, we use the term CHOX to represent all organic compounds identified by FIGAERO-CIMS, $C_{x\geq1}H_{y\geq1}O_{z\geq1}X_{0-n}$, detected as clustered with $I^-$, with X being different atoms including N, S, Cl, or a combination of them.

### 2.2.3 Background subtraction

The background in offline FIGAERO-CIMS is a combination of instrument background and field blank. The field blanks provide information on sampling and handling artefacts, while the instrument background is mainly from (1) the desorption of semi-volatile or low-volatile compounds adsorbed on instrument surfaces (such as the ion-molecular reaction region (IMR)), and (2) impurity of the reagent ion precursors and carrier gases. Thus, instrument background signal can vary for different samples and depending on instrument status. For FIGAERO-CIMS online deployments, frequent blank measurements and calibrations are recommended (Bannan et al., 2018; Thornton et al., 2020). The common method for online FIGAERO-CIMS of placing an additional filter upstream of the FIGAERO filter is impossible for offline pre-sampled filters. Given *1)* the large variation of the filter sample loadings (~$1\times10^{-2}$ µg –1.2 µg), which influences the number of compounds that can potentially adsorb to instrument surfaces, *2)* the general scarcity of field blanks in offline mode compared to background filter samples in online FIGAERO-CIMS, and *3)* that the instrument background can be influenced by instrument history very different from the offline sample due to the temporal separation of sample and analysis, choosing an appropriate instrumental and field blank determination method is crucial and challenging for offline FIGAERO-CIMS analysis. Here we describe and discuss performance of 6 different background subtraction methods (schematically shown in Fig. 2):

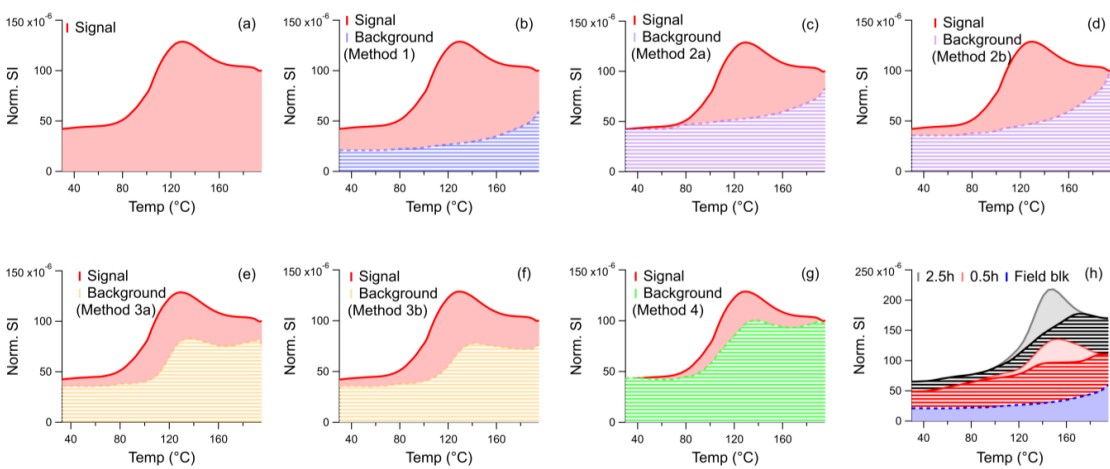

**Figure 2.** Schematic of a compound's signal and background thermograms for different background determination methods. The x-axis is the temperature during ramping, and the y-axis is the signal intensity normalized by the primary ion ($I^-$). (a) total sample signal of a model compound without blank subtraction, (b) Method 1: field blank only, (c) Method 2a: scaling field blank to the start of ramping, (d) Method 2b: scaling field blank to the end of soaking, (e) Method 3a: reheating of a subset of



filters, and using the average signal ratio from reheated and heated filters as background signal for all filters (individual
compound-based), (f) Method 3b: reheating of a subset of filters, and using an exponential fit to the entire mass range of the
average signal ratio from reheated and heated filters as background signal for all filters, (g) Method 4: thermal baseline using
a spline algorithm, and (h) one 0.5-h and one 2.5-h sample with blank-subtraction. Ideally, the *Is* of the 2.5-hour collection
sample ($Is_{2.5h}$) would be close to the sum of the 5 paralleled 0.5-hour collection sample ($Is_{0.5h}$).
**Method 1:** Background is the average integrated signal intensity (*Is,* the integrated signal of the thermograms
shown in Fig. 2a) of field blanks ($\overline{Is_{field\ blk,i}}$), which are three in our case (Fig. 2b). The integrated background-
subtracted signal for compound *i* ($Is_{blksub,i}$) is then $Is_i$ - $\overline{Is_{field\ blk,i}}$ .
**Method 2:** Background is field blank average ($\overline{Is_{field\ blk,i}}$, see Method 1) scaled to the ratio of ambient sample
and field blank signals during a reference period (ref period) – either prior to the start of heating (the first 1.5 to 3
min of the ramping procedure before the temperature starts to increase, Method 2a or at the end of the soaking (the
last 1.5 to 3 min of the soaking period, Method 2b). Method 2 corrects for variation in instrument background that
is not necessarily related to the sample to be analyzed. The integrated background-subtracted signal for compound
*i* ($Is_{blksub,i}$) is then
$$Is_{blksub,i} = \int I_{sample,ij} \ - \int I_{field\ blk,ij} \times \frac{\int^{ref\ period} Is_{i,ambient}}{\int^{ref\ period} Is_{i,field\ blk}} \qquad (1)$$

By using Method 2a, it is assumed that the signal measured before heating, but with the filter already in place, is
due to instrument background, which can vary between the measurement of a sample filter and a blank filter (Fig.
2c). However, this method may lead to underestimation of the sample signal for compounds that already evaporate
at room temperature.
By using Method 2b, it is assumed that the signal measured at the end of soaking is due to instrument background,
which can vary between the measurement of a sample filter and a blank filter. The variation in instrument
background is taken into account at maximum heating temperature (200 °C) and thus elevated temperature of
surfaces downstream of the filter, and at the end of the soaking period when presumably all material that can
evaporate from the filter has evaporated (shown in Fig. S1).
**Method 3:** In this method (Siegel et al., 2021), the instrument background is assessed by heating the same filter
twice, assuming that during the first heating cycle, all detectable material has evaporated, and that what is measured
in a reheating cycle is the instrument background signal. Ideally, reheating would be done for each sample
individually. Since this was not done for our dataset, the instrument background determined based on a few reheats
(3 in our case, the details of the reheating samples are shown in Table 1) had to be extrapolated to all samples
(Method 3a and 3b). It is clearly shown that the signals from the reheating cycle are much lower than those from
the first heating (Fig. S1) without a clear peak in thermograms for both filter types, suggesting sampled compounds
were well desorbed in the original heating cycle. Simple reheating does not consider the field blanks, which need
to be subtracted in addition.
For Method 3a we assumed that the ratio of the integrated signal of the second heating cycle (heating C2) and first
heating cycle (heating C1) of the same filter is influenced by volatility and therefore compound-dependent. Here
we used the average ratio from 3 reheating tests done for this dataset (Fig. S2). The distribution of the ratios is
shown in Fig. S3. The $Is_{blksub,i}$ was then calculated following Eq. 2, where the instrument background is the fraction
of the sample signal established from the re-heating, and added to the signal from the field blank, which is
calculated in the same way.
$$Is_{blksub,i} = \left( Is_{sample,i} - Is_{sample,i} \times Is_{i,\left(\frac{heating\ C2,i}{heating\ C1,i}\right)} \right)$$

$$- \left( Is_{field\ blk,i} - Is_{field\ blk,i} \times Is_{i,\left(\frac{heating\ C2,i}{heating\ C1,i}\right)} \right) \qquad (2)$$

For Method 3b, we assumed that the ratio of heating C2 to heating C1 exhibits a signal dependency (relatively
higher background for compounds with lower signal), calculated using an exponential fit to the data from the 3
reheat tests (Fig. S4) using Eq. (3) with the constants A, B, and C. The field blanks are calculated in the same way.
Then the $Is_{blksub}$ can be calculated as in Eq. (2)
$$Is_{i,\left(\frac{heating\ C2,i}{heating\ C1,i}\right)} = A + B \times \exp(Is_{sample,i} + C) \quad (3)$$

**Method 4:** Thermal baseline subtraction. In this method, we determined for every thermogram of each compound
a background thermogram termed thermal baseline ($Is_{thbsl}$). The thermal baseline was computed using a spline
algorithm initially developed by Wang et al. (2018) for determining the background concentration of a pollutant
using its concentration time series (by determining the spline of background from varying time intervals).
Thermogram data were pre-averaged to 1.8 mins (corresponding to 4 data points of the original time resolution of
27s) to reduce noise for the thermal baseline computation. Field blanks were handled in the same way. Thus, the
blank-subtracted signal $Is_{blksub}$ of a compound $i$ is:

$$Is_{blksub,i} = Is_{sample,blksub,i} - Is_{field\ blk,blksub,i}$$
$$= \left(\int I_{sample,i,j} - Is_{sample,thbsl,i}\right) - \left(\int I_{field\ blk,i,j} - Is_{field\ blk,thbsl,i}\right) \quad (4)$$

$Is_{sample,\ thsbl,i}$ and $Is_{field\ blk,\ thbsl,I}$ represent the thermal baseline of compound $i$ for samples and field blanks, respectively.
**2.2.4 Thermograms and $T_{max}$ recovery**
The amount of compounds coming off the filter at a certain temperature varies as a function of temperature
ramping rates, resulting in different thermogram shapes and $T_{max}$ (shown in Fig. 1d). This is especially important
in our case for the non-uniform ramping protocols. In an attempt to make the different cases comparable for
qualitative volatility studies, we developed a thermogram correction where the blank-subtracted signal as a
function of temperature for each compound $i$ is re-distributed to constant temperature intervals (Eq. (5)):
$$I_{thermocorrected,i,j} = \int_{T-\Delta t}^{T} I_{sample,blksub,i,j}\ dT \quad (5)$$

Considering the ~2 °C variation in thermogram reproducibility reported from an online FIGAERO-CIMS study
(Lopez-Hilfiker et al., 2014), the temperature interval $\Delta T$ used in this study is 3°C.

**3.  Results**
**3.1 Assessment of the background: Signal comparison between different blank subtraction methods**
To assess the influence of the 6 background methods on the resulting signal, Quartz fiber filter samples from 5
different 0.5-h samples (OA: ~2.0×10⁻² μg for each punch) and a 2.5 h sample collected in parallel (OA: 9.1×10⁻²
μg) were used, and the sum of their background-subtracted integrated signals ($Is_{blksub}$) compared (Fig.2 h). Without
background subtraction, the sum of the signals from the five 0.5-h samples was generally higher than the $Is$ of the
2.5-h sample (shown in Fig. 3a). An exception to this is $HNO_3^-$, which has the highest signal of all compounds and
therefore is the least influenced by background. The higher $Is$ for the sum of the five 0.5-h samples is likely because
of the low signal-to-noise ratio compared to the 2.5-h sample. Subtracting only the field blank (Method 1) therefore
yielded the same result (Fig. 3b). Scaling the heating baseline (Method 2a and 2b) led to a better agreement between
the sum of the five 0.5-h and the 2.5-h samples (Figs. 3c and d). Compounds with high abundance generally fall
on a 1:1 line (slope range 0.5–2) by using these two background subtraction methods. With the thermal baseline
subtraction method (Method 4), results were comparable between 2.5-h and five 0.5-h samples. For the approach





using filter reheating (Method 3), there was lesser agreement between the sum of the 0.5-h samples and the 2.5-h
sample (Figs. 3e and 3f). We speculate that this could be improved with a reheating cycle for every sample.
In general, as expected, high mass loadings are less sensitive to the various background subtraction methods due
to the higher signal-to-noise ratio (for example, 12-h/24-h sampling with OA loading of ~1 µg, Fig. S5). Besides
filter loadings, baseline levels can also be influenced by the properties of compounds (e.g. stickiness) and
instrument geometry. In summary, of all background subtraction methods shown here, Methods 2a, 2b, and 4
achieved the best agreement in signal intensities between the sum of 0.5-h and 2.5-h samples (Fig. S6). With these
methods, 82% to 93% of high-signal compounds (25% highest signal) fell into a signal ratio of ~1 (0–2). This
shows the importance of assessing the instrument background right, especially for compounds with low signal.

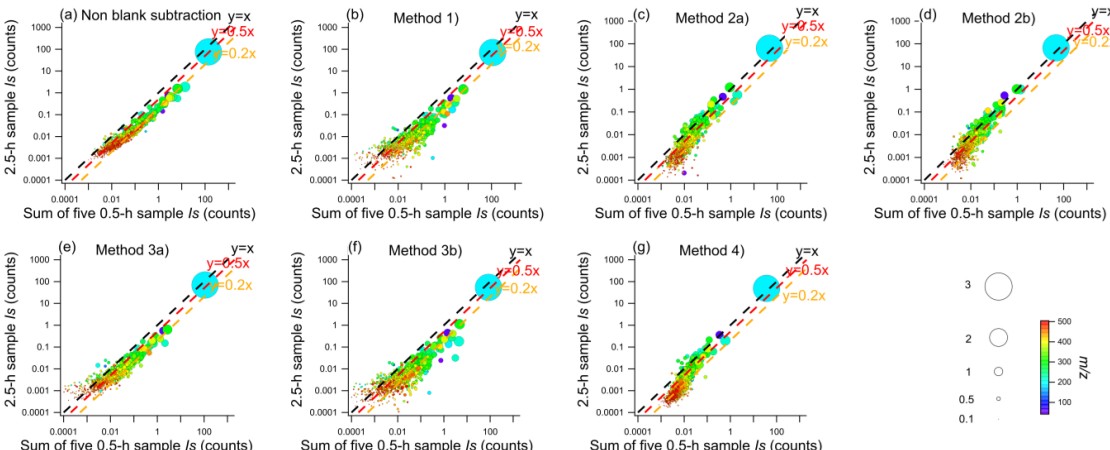


**Figure 3.** Comparison of the integrated signals ($Is$) for the 2.5-h versus sum of 0.5-h samples (a) without blank subtraction,
with blank subtraction using (b) Method 1, (c) Method 2a, (d) Method 2b, (e) Method 3a, (f) Method 3b, (g) Method 4. The
size of dots is proportional to the 4$^{th}$ root of integrated signal intensities of compounds, and they are color-coded by the ions'
$m/z$ (mass-to-charge ratio).

In this study, we applied Method 2b in the following discussions due to its better performance for the compounds
with both higher ($Is>0.1$ counts) and lower signal ($Is<0.01$ counts, Fig. 3d). First, we examined the signal-to-noise
ratios for offline FIGAERO-CIMS, defined as the ratio of the blank-subtracted signal to the standard deviation
(STDs) of the background determined using method 2b per compound. Most of the identified compounds are
above the estimated detection limit (3 times STDs of the backgrounds) for both filter types (87% and 87% of
CHOX peaks for both 24-h Quartz and Teflon filters, OA loadings of 1.2 µg/$3.1\times10^{-2}$ cm$^2$ (2 mm punch)). For the
12-h samples (OA loadings of 0.58 µg/$3.1\times10^{-2}$ cm$^2$ (2 mm punch)), 84% and 70% of CHOX compounds were
above the detection limit for Quartz and Teflon filters, respectively (Fig. S7). Evidently, this varies for different
filter loadings and punch areas.

### 3.2 Reproducibility of signal

We performed reproducibility tests using three 2-mm punches from the same 24-h and 2.5-h samples of both
Teflon and Quartz filters and checked the signal response with the non-uniform temperature ramping procedure.
The comparisons of the blank-subtracted CHOX $Is$ for the 24-h and 2.5-h sample punches for both filter types are
displayed in Fig. 4 and Fig. S8, respectively.
In Figs. 4a and 4b, we plotted the compounds' signal from one punch versus their average signal from all 3
punches for the Teflon and Quartz filters, respectively. We observe a high correlation between the individual and
average signals (Spearman correlation coefficients Rsp are 0.95–0.96 and 0.97–0.99 for Teflon and Quartz filters,





respectively). For each CHOX compound, we also computed the relative error (standard deviation/average signals
(Std($Is$)/Avg($Is$) for the three punches) versus the average signal (Figs. 4c, 4d). The relative error for a CHOX
compound was 9% for Quartz and 18% for Teflon (median relative errors) for 24-h samples (Figs. 4c, 4d). The
relative error decreased with higher signal intensities (Figs. 4c, 4d), especially for the Quartz filters, suggesting
that abundant compounds are measured more precisely than less abundant compounds. This trend is less apparent
for Teflon filters, which is likely caused by less reproducibility for high $Is$ compounds. Possible explanations could
be uneven distribution of particulate mass on the filter or larger uncertainties in the punching process for Teflon
filters due to the extension of the material. 86% and 94% of all CHOX compounds for Teflon and Quartz filters,
respectively, had >3 times higher signals than the variability from the duplicate tests (Fig. S7). For the 2.5-h filter
samples (Fig. S8), the relative error is higher compared to the 24-h samples (25% for Quartz, and 31% for Teflon).
This is likely due to the lower OA loadings ($9.1 \times 10^{-2}$ µg/punch) of the 2.5-h sample compared to the 24-h sample
(1.2 µg/punch), which leads to higher uncertainties for blank subtraction and peak fitting. Still, the analytical
reproducibility is acceptable, even for samples with OA loadings as low as ~0.1 µg. The relative error between
repeats reported here is slightly larger (~9% and 18% for ~1 µg OA/punch for Quartz and Teflon filters, and 25%
for Quartz, 31% for Teflon for ~0.1 µg OA/punch) compared to the variability in signal for online FIGAERO-
CIMS (5–10% for 1 µg OA, (Lopez-Hilfiker et al., 2014)).

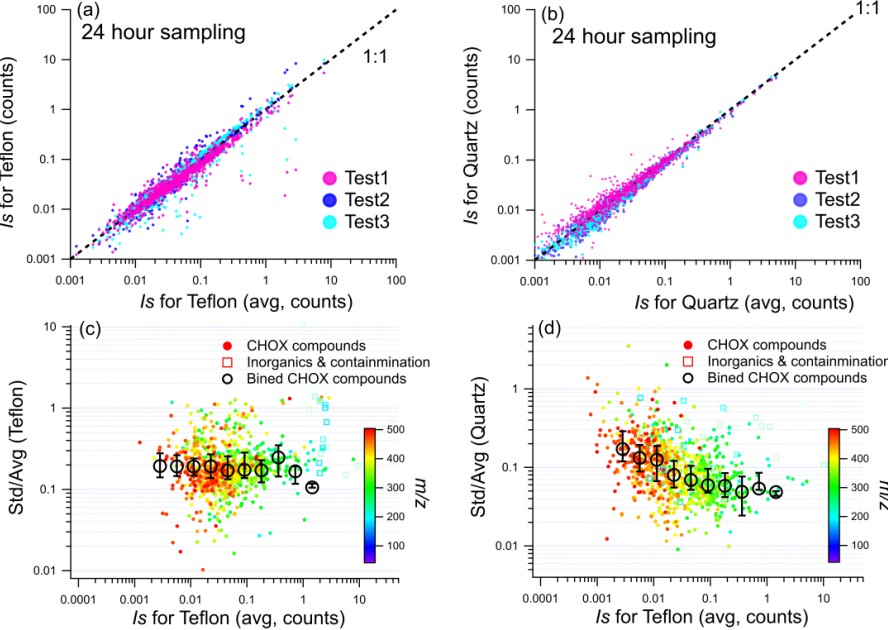


**Figure 4**. Comparison of the integrated signals from duplicate tests of the same 24-h sample for (a) Teflon and (b) Quartz
fiber filters. The relative error ($Is$ ratio of standard deviation/average) value of the 3 duplicate tests as a function of $Is$ for (d)
Teflon and (d) Quartz filters. In (c) and (d), CHOX compounds are shown as dots, inorganics as well as contaminants as
squares colored by the $m/z$. The black cycles in (c) and (d) represent median values of signal intensity bins (with log $Is$ intervals
of 0.3 for the $Is$ range of 0 to 2) and error bars represent the 25th and 75th percentile of binned values of Std($Is$)/Avg($Is$) for
CHOX.
**3.3 Comparison of signal for different temperature ramping protocols**
Here we compare the signal from different ramping protocols for the punches from the same 24-h Quartz and
Teflon filters (Table 1). Since as suggested in the section 2.2.2, the $Is$ were calculated by the integration of the





normalized signals (normalized to the primary ion (I⁻)), which to some extent compensates for reagent ion
depletion. The signal of the field blanks is largely dominated by instrument background (i.e. there is no distinct
peak in the thermogram (Fig. S1e) thus the $Is$ of the field blanks is highly influenced by integration time. Since
the field blanks were only analyzed with non-uniform ramping, the $Is$ for slow non-uniform and uniform ramping
protocols were assumed as the $Is$ of non-uniform scaled by their integration time ratios.
The comparison of the background-subtracted $Is$ of all identified compounds from different ramping protocols
for a pair of 24-h Quartz and Teflon filters each is shown in Fig. 5. Since the integrated signals of the compounds
within a mass spectrum are log-normally distributed (shown in Fig. S9a and 9b), a linear fit would be strongly
biased by high-signal compounds such as $HNO_3I^-$ or $C_6H_{10}O_5I^-$. Thus, we calculated the correlation coefficients of
the log-transformed signal intensities in the comparison. The Pearson correlation coefficients (Rp) and Spearman
correlation coefficients (Rsp) are as follows: for Quartz filters Rp = 0.91, Rsp = 0.94 for non-uniform $vs$ uniform,
and Rp = 0.91, Rsp = 0.94 for slow non-uniform $vs$ uniform, and for Teflon filters Rp = 0.82, Rsp = 0.78 for non-
uniform $vs$ uniform, and Rp = 0.83, Rsp = 0.70 for slow non-uniform $vs$ uniform protocols.
These numbers suggest that the Quartz samples were less affected by different temperature ramping protocols
than the Teflon samples. We also note that Teflon samples exhibited lower reproducibility than Quartz samples
(see section 3.2). The lowest Rp and Rsp were observed for the comparison between the slow non-uniform ramping
and the uniform ramping procedure for Teflon filters (Fig. 5d). Possible explanations could be the higher
background and thus lower signal-to-noise ratios for Teflon filters in the low ramping rate region (1.3 °C min⁻¹ for
the range of 60 °C to 105 °C) of the slow non-uniform ramping protocol. Thus, care needs to be taken when using
very slow heating rates and backgrounds need to be carefully assessed, especially for Teflon filters.

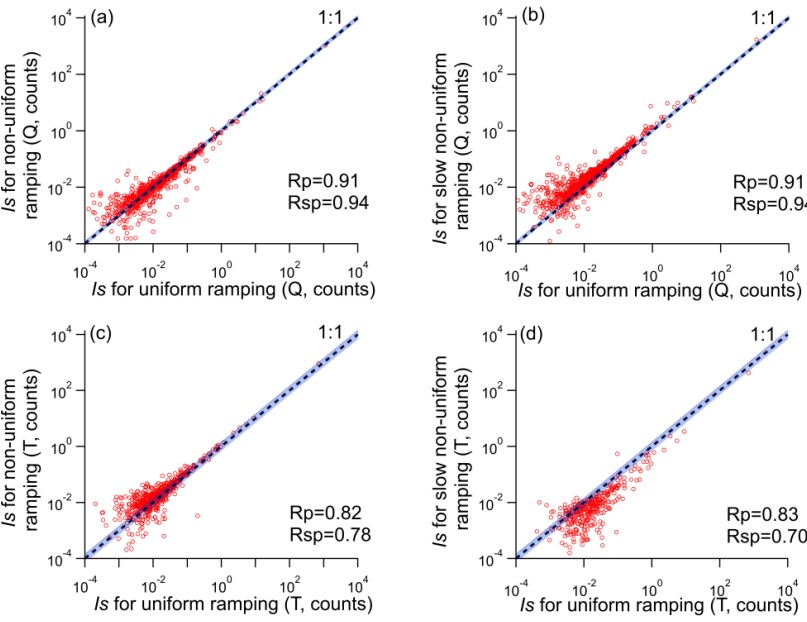


**Figure 5**. Comparison of $Is$ from the different temperature ramping protocols of the 24-h Quartz (Q) and Teflon (T) filter
samples, (a) non-uniform and uniform ramping (Quartz sample), (b) slow non-uniform and uniform ramping (Quartz sample),
(c) non-uniform and uniform ramping (Teflon sample), (d) slow non-uniform and uniform ramping (Teflon sample). The blue
shaded areas represent the relative error of signal assessed in the reproducibility tests of the 24-h samples (18% for Teflon
and 9% for Quartz filters). The upper and lower limits for the reproducibility-based variation are calculated as (1+18%)/(1-



18%) and (1-18%)/(1+18%), respectively. The upper and lower limits for the $Is$ distribution of Quartz caused by
reproducibility are calculated as (1+9%)/(1-9%) and (1-9%)/(1+9%), respectively.
For further analyses, we use the results from the non-uniform temperature ramping protocol, which represents a
good balance between the influence of background due to low signal-to-noise ratios, and $I^-$ depletion. The good
agreement between offline FIGAERO-CIMS and ToF-ACSM discussed in Section 3.5 further implies that such a
ramping protocol is suitable for the OA loadings observed in our study.

### 3.4 Linearity of signal response

To assess the linearity of signal response to the amount of sample collected on the filter, we used punches with
varying areas from one single filter. We used punch diameters of 2, 3, 4, and 7 mm for a Teflon filter and 2 mm
and 3 mm for a Quartz filter. The analytical protocol was kept constant between the individual sample punches
(non-uniform ramping protocol and method 2b for background subtraction). The mass loadings of the analyzed
filter punches ranged from 1.2 to 15 µg OA (2.2 to 27 µg $PM_{2.5}$) for the Teflon filter and from 1.2 to 2.7 µg OA
(2.2 to 5.0 µg $PM_{2.5}$) for the Quartz filter (Table 1). The blank-subtracted $Is$ from the different punching areas for
the Quartz and Teflon filters is shown in Fig. 6. Overall, the offline FIGAERO-CIMS approach responds linearly
to changes in filter mass loadings. The integrated signal ratios of CHOX are consistent with their respective area
ratios (Figs. 6a, 6b), within uncertainty. In Fig. 6c we also plot the signal ratios of the 2 mm punch to the other
punches, normalized by punching area (where 1 signifies perfect linearity). These ratios are generally in the range
of possible variability caused by the relative error from the reproducibility tests.

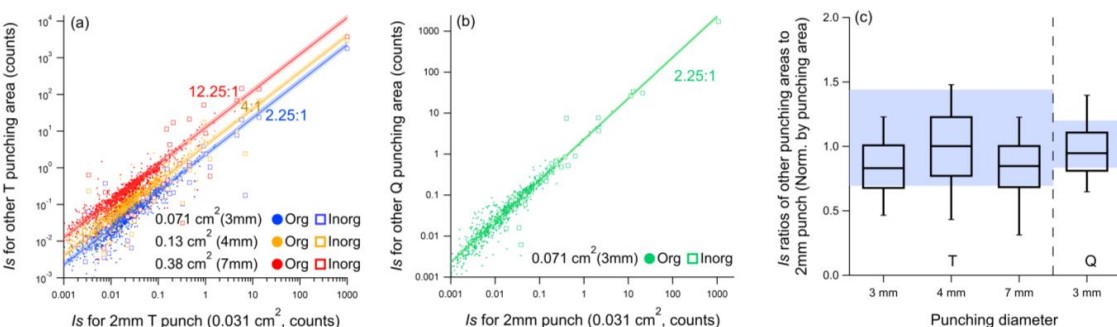

**Figure 6**. Comparison of the $Is$ between signals from punches (a) with 3 mm, 4 mm, 7 mm, and 2 mm in diameter for the
same Teflon filter, and (b) with 3mm and 2 mm in diameter for the same Quartz filter. The lines in (a) and (b) represent the
punching area ratios. The shaded areas in (a) and (b) represent the area ratio plus/minus the relative errors (9% for Quartz,
18% for Teflon) from the reproducibility tests. (c) Distribution of $Is$ ratios normalized by the punching area ratios (3 mm, 4
mm, and 7 mm to 2 mm diameter punches for Teflon, 3 mm to 2 mm diameter punches for Quartz). Within each box, the
median (middle horizontal line), 25th and 75th percentiles (lower and upper ends of the box), and 10th and 90th percentiles
(lower and upper whiskers) are shown. The shaded area in (c) represents the possible distribution of the $Is$ ratios due to the
relative error established from the 24-h sample reproducibility tests (18% for Teflon and 9% for Quartz filters). The upper
and lower limits for the Teflon $Is$ ratio distribution are calculated as (1+18%)/(1-18%) and (1-18%)/(1+18%), respectively.
The upper and lower limits for the Quartz $Is$ ratio distribution are calculated as (1+9%)/(1-9%) and (1-9%)/(1+9%),
respectively.

For compounds with very high signals, the response $Is$ ratio can deviate from the punch area ratio, not least also
due to the varying degree of reagent ion depletion. The highest $I^-$ depletions were ~35%, ~60%, ~68%, and ~70%
for 2mm, 3mm, 4mm, and 7mm punches, respectively. For e.g. the highest inorganic ($HNO_3I^-$) and organic
($C_6H_{10}O_5I^-$) ions, the $Is$ from a 7mm punch is only 30% and 67%, respectively, of what would be expected based
on punching area ratios (7mm to 2mm). For smaller punches (4 mm/3 mm), 75%/80% and 105%/107% of the



expected $HNO_3I^-$ and $C_6H_{10}O_5I^-$ signals, respectively, are detected. This indicates that for reduced amounts of
desorbing material provided by smaller filter fractions, the amount of reagent ion is sufficient during the whole
ramping process (lowest $I^-/C_6H_{10}O_5I^-$ signal ratio: ~$10^3$). In other words, if titration of reagent ion can be avoided
as much as possible (e.g. $I^-$/target ion signal ratio: ~$10^3$) the $Is$ responds linearly to concentration changes. In this
study, titration is non-apparent for OA loadings of <5 μg and $I^-$ signals of ~1 million. Therefore, it is recommended
to calculate OA loadings of the samples prior analysis to determine the punching sizes in offline FIGAERO-CIMS
analysis.
**3.5 Comparison between offline FIGAERO-CIMS and in-situ ToF-ACSM**
In the following, we compare the time series of the signals from offline FIGAERO-CIMS from Quartz filters and
the corresponding chemical components from online ToF-ACSM measurement. The comparison between the total
signal of all identified CHOX compounds and OA concentrations from the ToF-ACSM is displayed in Fig.7a.
Here, the FIGAERO-CIMS signals of five polyols ($C_8H_{18}O_5I^-$, $C_{10}H_{22}O_6I^-$, $C_{12}H_{26}O_7I^-$, $C_{14}H_{30}O_8I^-$, $C_{16}H_{34}O_9I^-$)
were excluded, which were contaminants from the lab due to their inexplicably high $Is$ in 3 of the 27 12-h samples
and the usage of diethylene glycol (DEG) in the lab. Even though $I^-$ is selective towards oxygenated organic
compounds, the total CHOX signal measured by offline FIGAERO-CIMS in this study highly correlates with OA
measured by the ToF-ACSM (Rp = 0.94), which is known to be dominated by secondary organic aerosols (SOA)
(Cai et al., 2020; Kulmala et al., 2021; Jia et al., 2008).
The time series of the 12h-$Is$ for $HNO_3I^-$ and $SO_3I^-$ measured by offline FIGAERO-CIMS correlate well with the
$NO_3$ and $SO_4$ concentrations from ToF-ACSM (Rp = 0.94 and 0.95, Fig. 7b). The signal of $HNO_3I^-$ in the particle
phase measured by FIGAERO-CIMS is as an indicator of particulate nitrate and organonitrate (Lee et al., 2016),
and the signal of $SO_3I^-$ is related to inorganic sulfate and sulfur-containing organics (Ye et al., 2021; Cao et al.,
2019). A similarly good correlation is observed between the signal intensity from the same offline FIGAERO-
CIMS method and $PM_{2.5}$ component concentrations measured in-situ by ToF-ACSM in a previous study conducted
in Beijing at Peking University campus (Zheng et al., 2021), which is shown in Fig. S10 (Zheng et al., 2021). The
generally good temporal correlation of different PM constituents between offline FIGAERO-CIMS and ToF-
ACSM analyses highlights the good performance of the offline FIGAERO-CIMS method, at least in terms of bulk
PM constituents.

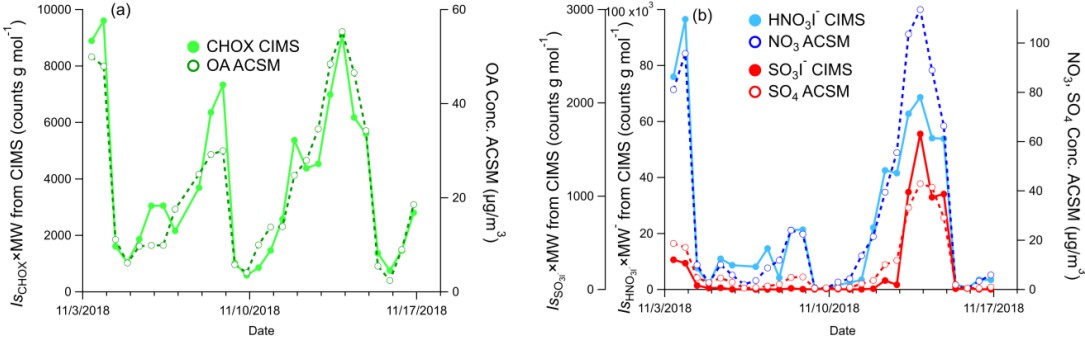


**Figure 7**. Comparison of the time series of the integrated signals of inorganic and organic compounds from 12-h samples (2
mm punches) analyzed by offline FIGAERO-CIMS, and chemical components measured in-situ by ToF-ACSM, (a) total
CHOX from FIGAERO-CIMS and OA from ToF-ACSM, (b) $HNO_3I^-$ from FIGAERO-CIMS and $NO_3$ from ToF-ACSM,
(c) $SO_3I^-$ from FIGAERO-CIMS and $SO_4$ from ToF-ACSM. To compare with the $PM_{2.5}$ component concentrations from the
ToF-ACSM, the $Is$ of each compound from FIGAERO-CIMS was multiplied by their molecular weight (MW) in (a) and (b).

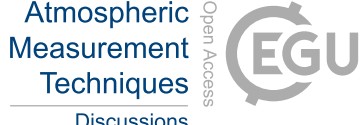

### 3.6 Comparison of Quartz and Teflon filters

In the following, we compare the *Is* from simultaneously collected Quartz and Teflon filter samples (collection times 2.5 h, 12 h, and 24 h, see Table 1). Fig. 8a and b show the comparison of the average *Is* of compounds (3 samples each) for both filter types, with 2.5h (OA loading of $9.1\times10^{-2}$ µg) and 24h (OA loading of 1.2 µg) collection times. The mass spectra show an overall similar pattern, we observe a non-negligible difference, especially for the 2.5h samples (Fig. 8a). The log-transformed signals from Quartz and Teflon samples correlate better for 24-h samples (Rp = 0.96, Rsp = 0.95, Fig. S9c) than for the 2.5-h samples (Rp = 0.88, Rsp = 0.87, Fig. S9d). In addition, the signal observed for Quartz filter samples is generally slightly lower than for Teflon filter samples (Fig. 8c, d). Compounds with high Quartz/Teflon-signal ratios are in general semi- or low volatile compounds (operationally defined as having a $T_{max}<60$ °C). These compounds tend to be in the CHO and especially CHON category and exhibit a higher degree of unsaturation (e.g. $C_8H_6O_3I^-$, $C_6H_5NO_3I^-$ and $C_7H_6NO_3I^-$). They can be aromatics or their thermal fragmentation products (Liu et al., 2019). Due to the high surface area of the Quartz filters, semi- or low volatile compounds are more easily adsorbed than on Teflon filters, potentially resulting in higher positive artefacts. Compounds with low Quartz/Teflon-signal ratios tend to have overall low signal. Despite the application of a blank determination method that takes instrument backgrounds into account (Method 2b), higher residuals were still observed for the lower signal compounds, especially for the Teflon filters (as seen also for the 2.5-h and 0.5-h sample comparison (Fig. 3d). In contrast, compounds with a higher signal tend to be in the range of Q/T ratios expected based on the observed variability from the reproducibility tests (shown in Fig. 8c and 8d).

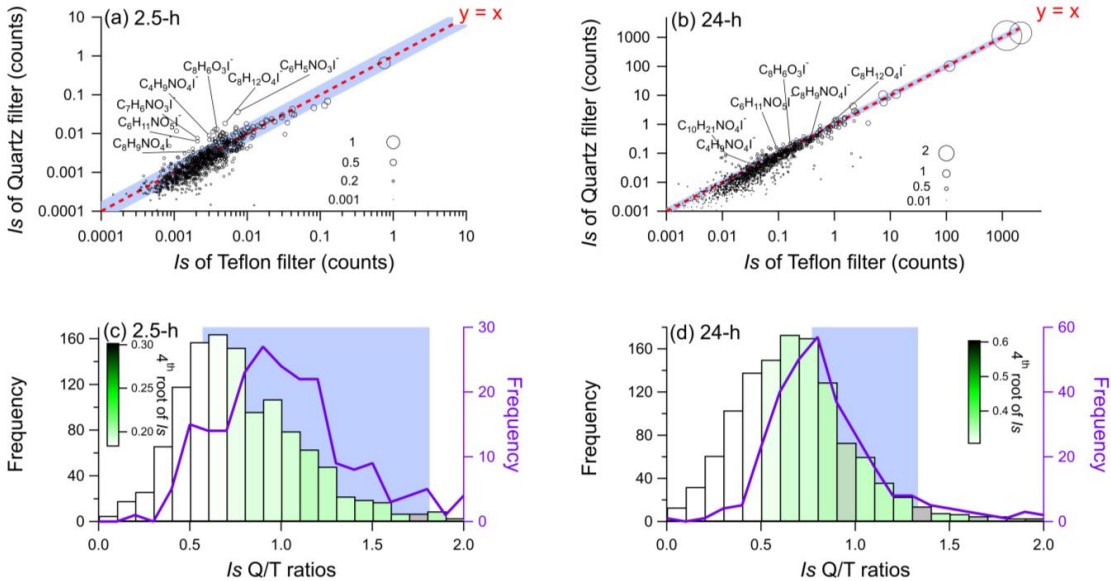

**Figure 8.** Comparison of the integrated signal intensities of all identified compounds for the Quartz fiber and Teflon filter samples, (a) 2.5-h samples, (b) 24-h samples. The size of symbols in (a) and (b) is proportional to the 4[th] root of the signal intensity of each compound from the Quartz filter. The distribution of *Is* ratios (green bars) of Quartz/Teflon, the distribution of *Is* ratios (purple lines) for the 25% of compounds with the highest signal for (c) 2.5-h samples, and (d) 24-h samples. The bars in (c) and (d) are colored by the average of the 4[th] root of the signal intensity for the Quartz filter. The shaded area in each panel represents the possible distribution of *Is* ratios of Quartz/Teflon from the relative errors from the duplicate tests of 2.5-h (25% for Quartz and 31% for Teflon) and 24-h (9% for Quartz and 18% for Teflon) samples. The upper and lower limits for the 2.5-h Quartz/Teflon *Is* ratios were calculated as (1+25%)/(1-31%) and (1-25%)/(1+31%), respectively. The upper and





lower limits for the 24-h Quartz/Teflon *Is* ratios were calculated as (1+9%)/(1-18%) and (1-9%)/(1+18%), respectively. The
$T_{max}$ was corrected.

### 3.7 $T_{max}$: Influence of temperature ramping protocol and filter type

Non-uniform ramping of the temperature due to reagent ion titration is more likely needed when the FIGAERO-
CIMS is run in offline mode compared to online mode, where sampling times and resulting filter mass loadings
can be adjusted more easily. We have therefore developed a method (see section 2.2.4) to recover $T_{max}$ from non-
uniform ramping protocols, i.e. to make it comparable to $T_{max}$ from uniform ramping protocols. Compared to the
raw thermograms, the shape of the corrected thermograms is more similar to that of the uniform protocol (Fig. S11
and S12), since the thermograms were re-gridded to the same temperature intervals (3 °C).
Firstly, we tested the variation of $T_{max}$ from the three duplicate tests of the Quartz filters using the non-uniform
ramping protocol and thermogram correction (Fig. 9a). After correction, the corrected $T_{max}$ ($T_{max\_nonuni\_corr}$) from
individual tests was highly correlated with their average ($T_{max\_corr\_avg}$, Rp = 0.87–0.93). The median value of the
difference between $T_{max\_nonuni\_corr}$ of duplicate tests and their average for all compounds ranges from -2.7–0.7 °C
(shown in Fig.9b). The majority of compounds (52%–70%) have a $T_{max}$ difference within 5 °C, close to the value
reported in previously (~2°C, (Lopez-Hilfiker et al., 2014)). The median standard deviation of the difference
between the corrected $T_{max}$ of individual tests ($T_{max\_nonuni\_corr}$) and their average ($T_{max\_corr\_avg}$) from all compounds
is 5.7 °C, which is defined as the variation of $T_{max}$ for duplicate tests.

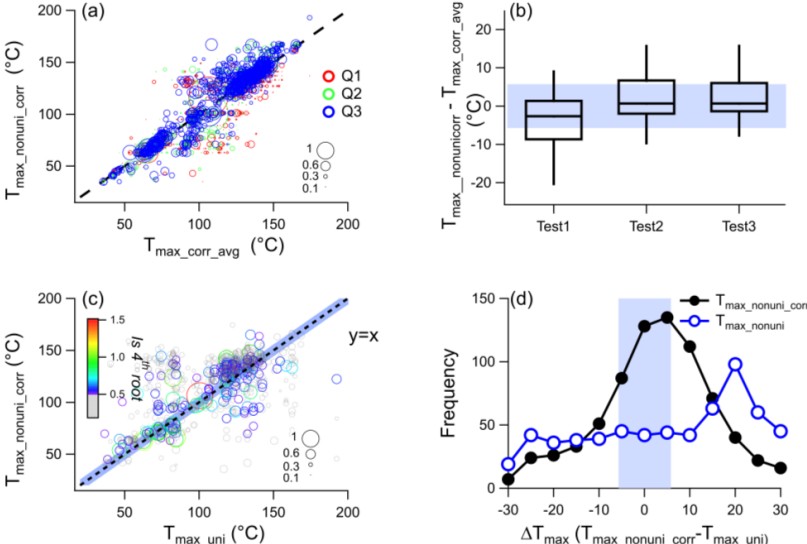


**Figure 9**. (a) Comparison of $T_{max\_nonuni\_corr}$ from the 3 duplicate tests and their average ($T_{max\_corr\_avg}$), (b) distribution of the
difference between the 3 triplicate tests and the $T_{max\_corr\_avg}$, (c) comparison of $T_{max}$ from the corrected non-uniform ramping
and uniform ramping protocol ($T_{max\_uni}$), (d) histogram of $\Delta T_{max}$ between $T_{max}$ from the uniform ramping protocol ($T_{max\_uni}$)
and non-uniform with ($T_{max\_nonuni\_corr}$)/without ($T_{max\_nonuni}$) correction. The size of symbols in (a) and (b) is proportional to the
$4^{th}$ root of the integrated signal intensity. The $4^{th}$ root of the signal intensity <0.5 is shown in grey. The uniform ramping
protocol test and 3 duplicate non-uniform ramping protocol tests were conducted for the same 24-h Quartz filter (Nov 23 to
24). The shaded area in (b), (c), and (d) represents $T_{max}$ variation (±5.7°C) from the duplicate tests.
We take the uniform sampling protocol (see Fig. 1d) as the basis since this is the commonly used protocol for
FIGAERO-CIMS in online mode. The comparison of $T_{max}$ from the corrected non-uniform and the uniform
ramping protocols is shown in Fig. 9c. Generally, after correction for the non-uniform ramping, the Pearson
correlation coefficient of $T_{max\_nonuni\_corr}$ and $T_{max\_uni}$ is higher (Rp = 0.60) compared to the uncorrected ones with





the uniform protocol (Rp = 0.20, $T_{max\_nonuni}$ *vs* $T_{max\_uni}$). The correlation coefficients were even higher (0.72 and
0.84) for the 400 and 100 compounds with the highest signal intensity. In Fig. 9d we plot the frequency distribution
of the differences between the corrected $T_{max}$ ($T_{max\_nonuni\_corr}$) and $T_{max}$ from the uniform protocol ($T_{max\_uni}$) for each
CHOX compound in the spectrum. For 73% of the compounds, the difference in $T_{max}$ between the two ramping
protocols lies between -15 and 15 °C, and 41 % of compounds exhibit a difference of 0 – ±5 °C.
In the next step, we compared the volatility derived from $T_{max}$ for Quartz fiber and Teflon filters. We selected a
number of inorganic and organic compounds, based on their high average signals for the whole sampling period,
for comparison of thermograms from 12-h and 24-h Teflon and Quartz filters sampled in parallel (Table S1, Fig.
10). Compounds include $HNO_3I^-$, CHON ($C_6H_5NO_3I^-$, $C_7H_7NO_3I^-$) and CHOS ($CH_4SO_3I^-$, $C_2H_4SO_4I^-$) compounds
as well as CHO compounds with $C_{num} \geq 3$ ($C_3H_4O_4I^-$, $C_4H_6O_4I^-$, $C_5H_8O_4I^-$, $C_6H_8O_4I^-$, $C_6H_{10}O_4I^-$, $C_6H_{10}O_5I^-$).
Compounds with $C_{num} < 3$ (e.g. $CH_2O_2I^-$) were excluded due to possible gas-phase interference and more likely
influenced by thermal decomposition. Some compounds exhibited similar thermogram shapes for the two types of
filters, such as $C_6H_{10}O_5I^-$ and $CH_4SO_3I^-$, while for some other species, the thermograms were different. Taking
$C_3H_4O_4I^-$ as an example, a bimodal thermogram shape with peaks around 100 °C and 150 °C was observed for the
Quartz filter, while only a unimodal peak around 90 °C was observed for the Teflon filter. The different
thermogram shapes of individual compounds for the different filter types might warrant further investigation with
a focus on the role of filter type properties (such as pore size, thickness, absorption, and hydrophobic/hydrophilic
properties).

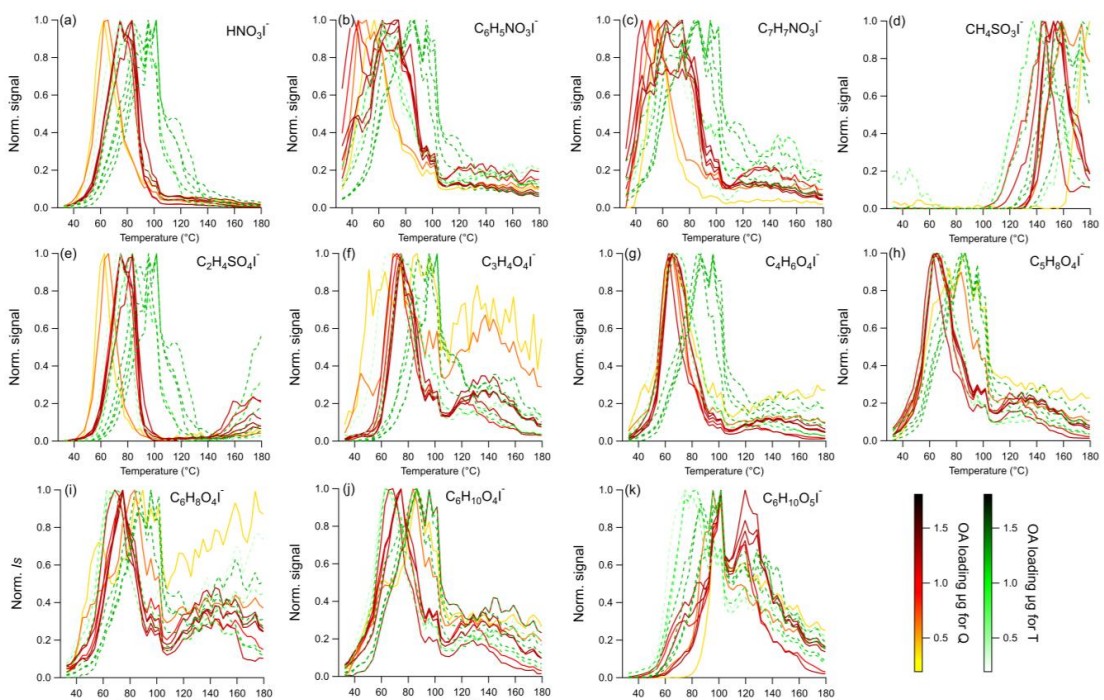


**Figure 10.** Normalized thermograms for Teflon (T, dashed lines) and Quartz (Q, solid lines) filters of, (a) $HNO_3I^-$, (b)
$C_6H_5NO_3I^-$, (c) $C_7H_7NO_3I^-$, (d) $CH_4SO_3I^-$, (e) $C_2H_4SO_4I^-$, (f) $C_3H_4O_4I^-$, (g) $C_4H_6O_4I^-$, (h) $C_5H_8O_4I^-$, (i) $C_6H_8O_4I^-$, (j) $C_6H_{10}O_4I^-$,
(k) $C_6H_{10}O_5I^-$. The thermograms were first corrected (section 2.2.4) and then normalized to signals in $T_{max}$ and colored by the
OA mass loading. The sampling information of the thermograms presented here is listed in Table S1.



In addition, we found that compounds with higher mass loadings appeared to have a higher $T_{max}$ (e.g. $C_2H_4SO_4I^-$
and $C_7H_7NO_3I^-$, shown in Fig 10), consistent with previous findings using Teflon filters (Huang et al., 2018;
Ylisirniö et al., 2021). The variability in $T_{max}$ induced by varying PM loadings is within 5°C for 29% of compounds,
and within 15°C for 54% of all compounds for Quartz filters, and 35% and 57% of compounds, respectively, for
Teflon samples. The $T_{max}$ variation due to filter type (Rp=0.27) is much larger than the one induced by filter
loadings. Thus, the direct comparison of $T_{max}$ between Quartz and Teflon filters is not feasible, warranting further
research.

## 4. Discussion

This study introduces methods and assesses the performance of using the FIGAERO-CIMS in offline mode, i.e.
to analyze particulate matter collected temporally and locally distant from the instrument on filter samples (Quartz
and Teflon). Such an approach greatly enhances the capabilities of the FIGAERO-CIMS for analyzing atmospheric
samples, as it enables the probing of the air at locations where and on occasions when *in-situ* deployments are
difficult.
Due to the difficulties in background determination for offline FIGAERO-CIMS, in this study, we propose
different background determination methods, which were further assessed by the comparison between samples
from 5 different 0.5-h samples and a 2.5-h sample collected in parallel. We applied non-uniform temperature
ramping to avoid reagent ion titration and a background scaling method taking interference of variable instrument
backgrounds into account. In general, the offline FIGAERO-CIMS approach using the methods presented in this
study can be used for providing OA composition information with typical offline sampling times (e.g. 12h and
24h) samples: (1) the reproducibility of integrated signal intensity is within ±20% for both filter types (18% for
Teflon and 9% for Quartz), (2) detected signals respond linearly to changes in the samples' mass loadings, (3) the
signals of CHOX and $SO_3I^-$, $HNO_3I^-$ correlated well with corresponding $PM_{2.5}$ chemical component concentrations
of OA, $SO_4$, and $NO_3$ measured by ToF-ACSM (Rp= 0.94 to 0.95), (4) the log-transformed mass spectra are highly
correlated (Rp>0.9) between Quartz and Teflon filters for typical offline sampling times (e.g. 12h and 24h), and
for high-signal compounds the *Is* ratios between Quartz and Teflon filters are generally within reproducibility
variation. Overall, this highlights the possibility of using widely available and stored Quartz filters to identify
CHOX molecular composition with FIGAERO-CIMS.
$T_{max}$ retrieved from corrected thermograms of desorption with non-uniform ramping protocols are comparable to
$T_{max}$ from uniform ramping protocol for high signal intensity compounds (Rp = 0.72–0.84). More than 50% of
compounds have $T_{max}$ values that are reproducible within 5 °C for duplicate tests (Rp = 0.87–0.93) of the same
sample, and for >50% of compounds, $T_{max}$ varies within 15 °C for different mass loadings. Yet, $T_{max}$ is strongly
affected by the filter material (Teflon *vs* Quartz) leading to a large discrepancy in $T_{max}$ between Quartz and Teflon
samples (Rp = 0.27), hindering direct comparisons and warranting further research.
In summary, using FIGAERO-CIMS to analyze offline samples is a useful and simple way to investigate OA
molecular composition, but care needs to be taken for $T_{max}$ analyses. This opens broad applications to study OA
molecular composition, sources, and formation processes at several sites simultaneously and in long-term
deployments.

*Author contributions*

JC, KRD, CM, and MK designed the research. JC, FXZ, and WD collected the samples at the BUCT site. JC, CW,
SH, KRD, and CM analyzed the samples and interpreted the data. ZY and CQ analyzed the samples collected at
the Peking University campus site. CM, KRD, and MK supervised this research. JC, KRD, and CM wrote the



manuscript with contributions from all co-authors. All authors have given approval to the final version of this
manuscript.

*Acknowledgements*
The work is supported by the Knut and Alice Wallenberg Foundation (WAF project CLOUDFORM, grant no.
2017.0165), the Academy of Finland (Center of Excellence in Atmospheric Sciences, project no. 307331, and
PROFI3 funding, 311932, ACCC Flagship 337549), the European Research Council via ATM-GTP (742206),
Wihuri Foundation, and the Jane and Aatos Erkko Foundation. KRD acknowledges support by the SNF mobility
grant P2EZP2_181599. The authors also would like to thank Federico Bianchi's kind help and suggestions as well
as the effort from all the researchers in the BUCT project to maintain the BUCT site.

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
