# Peer review of "Characterization of offline analysis of particulate matter with FIGAERO-CIMS"

_Atmospheric Measurement Techniques, 2022_

## Author Response (AR1)

**Reviewer #1**

In the response, referee comments are given in black, and our responses are given in blue. Changes made to the manuscript are marked in underlined blue. The line number is for the Revised Manuscript.

Cai et al. present a characterization of the ability of the FIGAERO-CIMS to sample aerosol collected offline and sampled by the instrument. The authors did an excellent job of creating the experiments and methods to investigate different aspects that would impact the overall quantification of the aerosol sampled from the filters. The paper was an enjoyable read and is an excellent paper for AMT. Below are comments for the authors to address to improve clarification and the paper.

Reply: We are very grateful for the positive comments and helpful suggestions. We have carefully revised our manuscript accordingly.

Major

(1) In the introduction, the authors have a paragraph stating, "Both online and offline techniques have their advantages and disadvantages and are associated with artefacts...". However, the authors then only discuss the advantages and disadvantages for online techniques. It would be useful for the authors to also briefly discuss the advantages and disadvantages for offline techniques, which also corresponds to some more comments below.

Reply: Thanks for the reviewer's comment. We add the following discussions for the offline techniques in the revised manuscript in Line 64 – Line 70:

"Both online and offline techniques have their advantages and disadvantages and are associated with artefacts (Turpin and Lim, 2001; Turpin et al., 2000). Offline techniques are an easy alternative to demanding online in-situ approaches requiring large human and financial resources. Moreover, one collected filter can be used for different analysis methods and purposes. However, the offline approaches are susceptible to sample handling and storage artefacts. The condensation and re-evaporation of vapors, and potential reactions on the filter during sampling and storage can result in both positive and negative sampling biases (Turpin et al., 2000; Cheng et al., 2009)."

(2) In Section 2.1, the authors mention that they are using a four-channel sampler. It seems that all four channels were being used for filters, but further clarification on how each channel was being used, and if the channels were sequential or parallel would be beneficial. Also, a discussion about any potential sizing effects from the different channels would be good.

Reply: To clarify the sampling process, the following description of the four-channel sampler has been added in Line 108 – Line 121:

Main text:

"The four parallel channels of the sampler had a shared $PM_{10}$ cyclone inlet and were equipped with 4 independent $PM_{2.5}$ cyclones and auto flow controllers for each channel. All channels were measuring the same size range of particles. A sizing effect from the interactions between different channels can therefore be neglected. The setup of filter type for each channel was as follows: Channel 1, Teflon (12 h for or 0.5 h); Channel 2, Quartz (12 h or 0.5 h); Channel 3, Teflon (24 h or 2.5 h); Channel 4, Quartz filters (24 h or 2.5 h). This is listed in Table 1. The flow rate was regularly calibrated individually for each channel during the sampling process.

To investigate the influence of filter mass loadings and collection time on the signal response, the following filter samples were taken: (1) 5 pairs of samples (Teflon/Quartz fiber filters, Channels 1 and 2) with 30 min deposition time on Dec 15, 2018 between 14:00 to 16:30 (Table 1). At the same time, an additional pair of Teflon/Quartz samples was collected for 2.5 hours using the other two separate

channels of the sampler (Channels 3 and 4). (2) 12-h samples of Quartz/Teflon filters (Channels 1 and 2) from Oct 26 to Oct 30 and Nov 3 to Nov 24 (here only the Quartz filters from Nov 3 to Nov 16 were analyzed (in total 27 pairs of samples), shown in Table 1). (3) 24-h Quartz/Teflon samples (Channels 3 and 4) from Oct 26 to Oct 30 and Nov 3 to Nov 25 (here only one pair of Teflon/Quartz filters was analyzed, shown in Table 1)."

(3) In Section 2.1 and others, it is unclear if any artefacts with using filters were investigated -- e.g., uptake of gases, evaporation of aerosol, chemistry on the filters, loss or changing of sampling during storage, loss or changing of sampling in filter preparation. A brief discussion concerning any of these artefacts would be beneficial in understanding this technique for quantification.

Reply: Artefacts during sampling and storage as mentioned by the reviewer cannot be fully excluded, similar to any other sampling technique. Here we can use the ACSM as a reference method with the quantified OA concentration ($r$ = 0.94). We add the following discussion on potential artefacts as suggested (Line 509 – Line 514):

Main text:

"Like other offline sampling methods, the offline FIGAERO-CIMS method may be affected by artefacts from sampling and storage of the filters. Both positive (absorption of gaseous OA), and negative artefacts (volatilization of collected OA), may occur during the sampling and storage, even if filters were stored frozen (Cheng et al., 2009). However, the signals from FIGAERO-CIMS correlate generally well with major components measured by TOF-ACSM, suggesting that those artefacts can be considered minor in our study, at least in terms of bulk PM constituents (Figure 7)."

(4) In Table 1 and throughout the text, the authors state the amount of OA loading per area punched. It is unclear how the authors quantified this number.

Reply: The OA loading on the entire filter was assumed to be homogeneously distributed and determined by the co-located TOF-ACSM. We used the flow volume of the offline sampler, corresponding OA concentrations from the TOF-ACSM, and the area ratio of the punch and the entire filter. For clarification, we add the following discussion in the Method section, Line 137 – Line 140:

"The OA loading on each filter ($OA_{filter}$) was determined relying on the OA concentrations from the co-located TOF-ACSM ($OA_{ACSM}$), the offline filter sample flow rate (16.7 L min$^{-1}$), the sampling time, the surface of the entire offline filter sample ($A_{filter}$), and the analyzed offline filter sample ($A_{punch}$) (Equation 1):

$$OA_{filter} = \frac{A_{punch}}{A_{filter}} \times OA_{ACSM} \times Sampling\ flow\ rate \times Sampling\ time \qquad (1)$$

"

(5) Section 2.2.1.3: With the FIGAERO-CIMS, it has been acknowledged that the ramping process used to sample the aerosol leads to some degradation of the aerosol. A discussion on how the different ramping protocols may impact the evaporation/degradation would be beneficial.

Reply: As discussed in previous studies with FIGAERO-CIMS (Thornton et al., 2020) and correctly stated by the reviewer, different ramping rates affect thermal fragmentation, signal-to-noise ratios of compounds and primary ion depletion, and titration. In order to avoid the latter, we used a non-uniform ramping protocol in this study, and discuss the resulting signal in the paper in detail. Overall, the applied ramp rates, however, are within the ranges of ramp rates used in online FIGAERO (Yang et al., 2021; Thornton et al., 2020), and a detailed discussion on artefacts related to the thermal desorption process is outside the scope of this study.

(6) Fig. 3. With the scales being log-log, it's hard to understand/appreciate the differences and which method is best for blank subtraction. Also, the eye is drawn to the low signal/high m/z data, where most of it falls below the 1:1 line for many of the methods. How important is that for the overall quantification?

Reply: In the manuscript, we used the log-log scales since the integrated signals for 2.5-h and 0.5-h filters vary over 4 orders of magnitude. Following the reviewer's comment, we also added the comparison on a linear scale in the revised Figure S7 in the Supplementary Information for reference. In the revised Figure S7, we used a linear scale from 0 to 0.2 for the low-signal compounds (>90% of the number of compounds) and a log scale >0.2. It clearly shows that the performance of Methods 2a), 2b), and 4) is much better than that of the other background subtraction methods as stated in the manuscript.

In order to determine the best blank subtraction method, we first compare the integrated signals for the 2.5-h versus the sum of five 0.5-h samples. The OA loadings on the 2.5-h and 0.5-h filters are much lower than those on typical offline filters with sampling periods of 12 h or 24 h. The compounds with very low signal intensity are assumed to be more influenced by the background and have higher uncertainty due to their low signal-to-noise ratios, which is validated by their higher relative error ($Is$ ratio of standard deviation/average) shown in Figure 4. Therefore, these low-signal compounds are not as indicative for the method selection as the high-signal compounds.

Moreover, after appropriate background corrections, under typical offline sampling times (12 h or 24 h), high linearity of signal response could be achieved for the majority of compounds (even with $Is$ < 0.01, shown in Figure 4 and Figure 6), suggesting a good overall performance.

[Figure]

**Figure R1 (Figure S7).** Comparison of the integrated signals ($Is$) of all compounds for the 2.5-h versus the sum of signals of five 0.5-h samples (a) without blank subtraction, with blank subtraction using (b) Method 1, (c) Method 2a, (d) Method 2b, (e) Method 3a, (f) Method 3b, (g) Method 4. The size of dots is proportional to the 4th root of integrated signal intensities of compounds, and they are color-coded by the ions' m/z (mass-to-charge ratio). Compounds with $Is$<0.2 are shown on a linear scale and compounds with $Is$>0.2 on a log scale

(7) Check the axis labels for Fig. 4 and Fig. 6. It appears either something is missing or the names were mixed.

Reply: The reviewer is right. The axis labels for Fig. 4 and Fig. 6 have been corrected. The revised Figures are as follows:

[Figure]

**Figure R2 (Figure 4)**. Comparison of the integrated signals from three individual duplicate tests of the same 24-h sample to their average for (a) Teflon and (b) Quartz fiber filters. The relative error ($Is$ ratio of standard deviation/average) value of the 3 duplicate tests as a function of $Is$ for (d) Teflon and (d) Quartz filters. In (c) and (d), CHOX compounds are shown as dots, inorganics as well as contaminants as squares colored by $m/z$. The black circles in (c) and (d) represent median values of signal intensity bins (with log Is intervals of 0.3 for the Is range of 0 to 2) and error bars represent the 25th and 75th percentile of binned values of Std($Is$)/Avg($Is$) for CHOX.

[Figure]

**Figure R3 (Figure 6)**. Comparison of the Is between signals from punches (a) with 3 mm, 4 mm, 7 mm, and 2 mm in diameter for the same Teflon (T) filter, and (b) with 3mm and 2 mm in diameter for the same Quartz (Q) filter. The lines in (a) and (b) represent the punching area ratios. The shaded areas in (a) and (b) represent the area ratio plus/minus the relative errors (9% for Quartz, and 18% for Teflon) from the reproducibility tests. (c) Distribution of Is ratios normalized by the punching area ratios (3 mm, 4 mm, and 7 mm to 2 mm diameter punches for Teflon, 3 mm to 2 mm diameter punches for Quartz). Within each box, the median (middle horizontal line), 25th and 75th percentiles (lower and upper ends of the box), and 10th and 90th percentiles (lower and upper whiskers) are shown. The shaded area in (c) represents the possible distribution of the Is ratios due to the relative error established from the 24-h sample reproducibility tests (18% for Teflon and 9% for Quartz filters). The upper and lower limits for the Teflon $Is$ ratio distribution are calculated as (1+18%)/(1-18%) and (1-18%)/(1+18%), respectively. The upper and lower limits for the Quartz Is ratio distribution are calculated as (1+9%)/(1-9%) and (1-9%)/(1+9%), respectively.

(8) In Sect. 3.5, please state what is being compared explicitly (signal from CIMS vs mass concentration from ACSM). Looking at the figure, it takes a bit to understand the axis are different for the two measurements, leading the reader to try to understand how the CIMS appears to have more mass than ACSM and/or the agreement changes.

Reply: We add the following information on how the molecular weight weighted signals were calculated in the revised manuscript. (Line 494 – Line 500)

"We compute the sum of integrated signals ($Is$, signal integration over the entire thermogram, counts) multiplied by their molecular weight (MW, g mol$^{-1}$) of all compounds from FIGAERO-CIMS for comparison to the corresponding PM$_{2.5}$ component concentrations from the ToF-ACSM. Even though I$^-$ is selective towards oxygenated organic compounds, the total MW-weighted CHOX signal measured by offline FIGAERO-CIMS in this study highly correlates with OA from the ToF-ACSM (Rp = 0.94), which is known to be dominated by secondary organic aerosols (SOA) (Cai et al., 2020; Kulmala et al., 2021; Jia et al., 2008)."

We also add the following note in the figure caption of Figure 7

"Note that FIGAERO-CIMS and ToF-ACSM data are on different axes"

(9) Fig. 8, label (c) and (d) y-axes with what each frequency corresponds to. It is very unclear what is being plotted by just looking at the figures. In general, all figure axes and/or figure panels should be label more explicitly to better understand what is being plotted.

Reply: The frequency in Fig.8 (c) and (d) represents the number of compounds in each $Is$ ratio bin of Quartz and Teflon filters. To explain the frequency histograms more explicitly, we revised the axis labels and figure captions in Fig.8. Similarly, the labels are also revised in Figure S8 and Figure S12 to make them more explicit.

[Figure]

**Figure R4 (revised Figure 8).** Comparison of the integrated signal intensities of all identified compounds for the Quartz fiber and Teflon filter samples for (a) 2.5-h samples, and (b) 24-h samples. The size of symbols in (a) and (b) is proportional to the 4$^{th}$ root of the signal intensity of each compound from the Quartz filter. Frequency distribution (number of compounds) per signal ratio of Quartz/Teflon for all compounds (green bars), and high-signal compounds (highest 25% signal compounds) only (purple lines) for 2.5-h samples (c), and 24-h samples (d). The bars in (c) and (d) are colored by the average of the 4$^{th}$ root of the signal intensity of the Quartz filter. The blue shaded area in each panel represents the possible distribution of $Is$ ratios of Quartz/Teflon from the relative errors from the

duplicate tests of 2.5-h (25% for Quartz and 31% for Teflon) and 24-h (9% for Quartz and 18% for Teflon) samples. The upper and lower limits for the 2.5-h Quartz/Teflon $Is$ ratios were calculated as (1+25%)/(1-31%) and (1-25%)/(1+31%), respectively. The upper and lower limits for the 24-h Quartz/Teflon Is ratios were calculated as (1+9%)/(1-18%) and (1-9%)/(1+18%), respectively.

(10) Something that is missing overall from this paper is what is the ultimate goal of this paper. It is expected that researchers use this method for quantitative information about aerosol or qualitative information about the aerosol? If quantitative, see point (3) above, but there are other aspects that need to be discussed, including but not limited to: (a) percent recovery from filter, (b) more explicit intercomparisons with online measurements (e.g., FIGARO co-located with sampling for direct comparison of what's being observed, how much, and any potential changes of the aerosol prior to offline sampling), and (c) calibrations. For point (c), though the main paper does not show any data in mass concentration, one figure in the SI (Fig. S10) has converted the FIGARO data from signal to ug m-3.

Reply: In this study, we focused on introducing the method and best practices for using the FIGAERO-CIMS in offline mode, which enables the probing of the air of locations where and on occasions when *in-situ* deployments of the FIGAERO-CIMS are difficult. We note, however, that it is not the scope of this paper to discuss aspects of offline FIGAERO-CIMS that also apply to its online deployment, such as e.g. general percentage of recovery from the filter or calibrations, as mentioned by the reviewer. Here in this manuscript, we propose a series of approaches to the FIGAERO-CIMS use in offline mode, which includes a "sandwich" sample preparation, a non-uniform temperature ramping protocol due to higher mass loadings compared to online use, thermogram correction methods, and background determination methods. Following our established best practices, we analyzed an ambient dataset from Beijing, China, where we identified ~1000 organic aerosol (OA) molecules and the time series of their signals. The time series of the sum of signals of all organic compounds correlated well with the OA concentrations measured by ToF-ACSM, validating the robustness of the offline FIGAERO-CIMS analyses – at least in terms of bulk PM constituents. While the quantification of each single chemical component is a pressing question given the rise of soft ionization mass spectrometry, this is not the focus of the current manuscript.

Coauthors of this study were also involved in another study using FIGAERO-CIMS in offline mode, where calibrations for a series of the chemical were performed (Zheng et al., 2021). We presented their results in Figure S12 as an indicator to show that the offline FIGAERO-CIMS method has the potential to be quantitative with proper calibrations (Zheng et al., 2021).

To avoid potential misunderstanding and make the discussions more explicit, we add the following expression in the introduction:

Line 95 – Line 97

"The potential to broaden its application for OA component measurements in future research is also discussed. We note, however, that it is not the scope of this paper to discuss aspects of offline FIGAERO-CIMS that also apply to its online deployment, such as e.g. general percentage of recovery from the filter or calibrations."

And we rephrased the corresponding discussions and the caption of Figure S12.

Main text Line 505 – Line 508

From:

"A similarly good correlation is observed between the signal intensity from the same offline FIGAERO-CIMS method and PM$_{2.5}$ component concentrations measured *in-situ* by ToF-ACSM in a previous study conducted in Beijing at Peking University campus, which is shown in Fig. S10 (Zheng et al., 2021)"

To:

"Following the same method, after calibrations, the quantified CHOX mass concentrations of offline FIGAERO-CIMS were found to be highly correlated with OA and SOA from ToF-ACSM in another dataset at the Peking University campus (PKU) in Beijing, indicating offline FIGAERO-CIMS analysis can be quantitative with proper calibrations (shown in Fig. S12 (Zheng et al., 2021))."

[Figure]

**Figure R5 (Figure S12).** Comparison between CHOX mass concentrations from FIGAERO-CIMS, organic aerosols (OA), and secondary organic aerosols (SOA) derived from ToF-ACSM at the Peking University Campus (PKU) site. The details of the site, the comparison setup, calibrations, and calculations can be found in Zheng et al. (2021). In total, CHOX accounts for about 32–60% of SOA measured by the TOF-ACSM in their study.

(11) Fig. S6. It is currently unclear how to interpret this figure. The authors stated that Method 2a, 2b, and 4 provide the most reliable/reproducible answer; however, if the value should be a normal distribution around 1, it appears that Method 1, 2b, 3a, and 3b would be the methods to select. Also, it is surprising that there are no negative values. A distribution of what is expected maybe valuable in this figure to compare to which method is working as expected.

Reply: In the original Fig. S6, we used all fitted peaks in the histogram to show the distribution of ratios of *Is* of the 2.5-h and the sum of the five 0.5-h filter samples. However, as discussed in Major Comment 6, the 0.5-h sampling time is much lower than the typical offline sampling time (12 h or 24 h). The compounds with a very low signal-to-noise ratio of the 0.5-h sample can bias the method comparison due to their higher uncertainties. We therefore only keep the compounds with the highest 25% of signal in the revised Fig. S8, which clearly shows that Methods 2a, 2b, and 4 have better performance than others and can be used the future offline FIGAERO-CIMS studies. The negative values of *Is* ratios are also added in the revised Figure 6. In the figure caption, we state clearly the percentage of the dataset included in the distribution plot.

[Figure]

**Figure R6 (revised Figure S8).** The distribution of *Is* ratios between the 2.5-h and the sum of five 0.5-h samples for the 25% of compounds with the highest signal intensity for different background subtraction methods. The distribution range is from -1 to 6 with bins of 0.5, which covers 82%, 61%, 94%, 93%, 90%, 72%, and 96% of the top 25% of compounds with respect to signal for no blank subtraction, Method 1, 2a, 2b, 3a, 3b, and 4, respectively.

Minor

(1) In Section 2.2.1.2, please check the sequential number in lines 147 - 153, as (3) is repeated twice.

Reply: Line 167 to 173 has been revised with the right sequence of numbers.

"The OA mass loadings of the filter punches were estimated with the co-located ToF-ACSM in this study (details shown in Table 1). To test the performance of the method, we did the following tests (Fig. 1, Table 1): (1) reheating a few filters to determine backgrounds (see section 2.2.4), (2) assessing different background subtraction methods, (3) reproducibility of signals from the same filter (section 3.4), (4) linearity of signal response from different punching areas from the same filter (section 3.4), (5) comparing signals from different ramping protocols (section 2.2.1.3), (6) comparison between offline FIGAERO-CIMS and online ToF-ACSM (section 3.5), (7) signals from different filter types (section 3.6), and (8) thermograms from different types of filters (section 3.7)."

(2) As the other methods have examples in the SI, showing an example of Method 4 would be beneficial.

Reply: We have added the following figure (Figure S5) in the revised SI as an example of the baseline calculation with Method 4.

[Figure]

**Figure R7 (revised Figure S5).** Thermograms for $C_6H_{10}O_5I^-$ of sample and field blank, and the thermal baselines for sample and blanks using background subtraction Method 4.

In the main text (Line 304):

"Field blanks were handled in the same way (shown in Fig. S5):"

We also revised the schematic plot for blank subtraction using Method 4 (Fig.2 g) according to that:

[Figure]

**Figure R8 (revised Figure 2(g))** Method 4: thermal baseline using a spline algorithm

(3) Line 312, please change "background right" to "background correctly"

Reply: Line 350 has been revised as follows:

"This shows the importance of correctly assessing the instrument background, especially for compounds with low signal."

(4) Fig. 3, the y=x, y=0.5x, and y=0.2x are hard to read and to understand that they refer to.

Reply: We removed the y=x, y=0.5x, and y=0.2x lines in the revised manuscript and added the comparison on a linear scale in the revised SI (shown in Fig. R1).

(5) Line 325, "Evidently" is not the correct word choice. Just start the sentence with "This"

Reply: Line 372 has been revised as suggested.

"This varies for different filter loadings and punch areas."

(6) Fig. 10, try to select different colorbars as the red/green leads to issues for color blind.

Reply: The color bars have been changed as suggested.

[Figure]

**Figure R9 (revised Figure 10).** Normalized thermograms for Teflon (T, dashed lines) and Quartz (Q, solid lines) filters of, (a) $HNO_3I^-$, (b) $C_6H_5NO_3I^-$, (c) $C_7H_7NO_3I^-$, (d) $CH_4SO_3I^-$, (e) $C_2H_4SO_4I^-$, (f) $C_3H_4O_4I^-$, (g) $C_4H_6O_4I^-$, (h) $C_5H_8O_4I^-$, (i) $C_6H_8O_4I^-$, (j) $C_6H_{10}O_4I^-$, (k) $C_6H_{10}O_5I^-$. The thermograms were first corrected (section 2.2.4) and then normalized to signals in $T_{max}$ and colored by the OA mass loading. The sampling information of the thermograms presented here is listed in Table S1.

**Reviewer #2**

In the response, referee comments are given in black, and our responses are given in blue. Changes made to the manuscript are marked in underlined blue. The line number is for the Revised Manuscript.

This manuscript describes and characterizes the use of the FIGAERO-CIMS to analyze particle composition via off-line filter analysis (i.e. filters that are collected outside of the instrument and later inserted in the instrument for analysis). This technique can enable FIGAERO-CIMS analysis of particle composition in a greater number and variety of environments as it does not require moving the instrument. The manuscript is well written and of interest to readers of AMT. I suggest publication of the manuscript after my comments below have been addressed.

Reply: We are very grateful for the positive comments and helpful suggestions. We have carefully revised our manuscript accordingly.

Major comments:

Correction and analysis methods: The manuscript is generally unclear on which methods are suggested for future use. For example, the authors conducted background correction in six different ways (e.g. Fig 2), and conducted some analyses to decide which corrections were most consistent with their data. For future work do they recommend that others also correct the background in six different ways? Or are the insights from their analyses sufficient to recommend a subset of methods for future use?

Reply: We thank the reviewer for pointing out this shortcoming. It was our aim with this study to present an overview of various background determination methods and to assess their performance in order for future users of this method to be guided by our analyses. We have added the following discussion in the revised manuscript to clarify this aim:

Line 334 – Line 343

"With the thermal baseline subtraction method (Method 4), results were comparable between 2.5-h and five 0.5-h samples. For the approach using filter reheating (Method 3), there was a lesser agreement between the sum of the 0.5-h samples and the 2.5-h sample (Figs. 3e and 3f). We speculate that this could be improved with a reheating cycle for every sample. For future offline FIGAERO-CIMS analyses, we recommend carefully determining the background. Following our assessment of blank determination methods, we suggest regular collections of field blanks and scaling their signal (Methods 2a/b), and if field blanks are not available, computing a thermal baseline (Method 4). If using the reheating approach as in a previous study with FIGAERO-CIMS in offline mode (Siegel et al., 2021), the background should be determined by conducting reheating desorption cycles for each sample and blank individually."

Reagent ion depletion: The authors mention that reagent ion depletion is not desired (e.g. line 155). It was unclear to me from reading the manuscript whether and how they corrected for reagent ion depletion (e.g. by dividing the analyte signal by reagent ion concentration)?

Reply: Reagent in depletion is indeed not desired, as we state in the manuscript. We, therefore, present several measures in this manuscript to reduce reagent ion depletion:

1) We decreased the sample loading by punching only a small area ($d$=2mm) from the whole sampled filter and then use the "sandwich technique" to analyze with FIGAERO-CIMS. Therefore, the mass loadings and reagent ion depletion can be greatly reduced. (Line 159 – Line 166)

2) We applied a non-uniform ramping protocol in order to reduce reagent ion depletion between 60 °C to 105 °C desorption temperature, where $HNO_3$ exhibits a maximum signal. (Line 187 – Line 193)

Normalization of signal to reagent ion is normally done for FIGAERO-CIMS data regardless of reagent ion depletion.

We now added more explanation to the normalization process in this study:

Line 224:

"The total signal of a compound per filter sample, defined as the integrated signals (*Is*), calculated by first normalizing by the signals of the primary ions (I⁻) and then integrating the entire thermogram (ramping and soaking)."

Data from the FIGAERO-CIMS and the ACSM are found to correlate well (e.g. Fig. 7). Do the authors have any information about their quantitative agreement?

Reply: We have added more information regarding the quantification of offline FIGAERO-CIMS to the revised manuscript (Line 505 – Line 508) and SI:

"Following the same method, after calibrations, the quantified CHOX mass concentrations of offline FIGAERO-CIMS were found to be highly correlated with OA and SOA from ToF-ACSM in another dataset at the Peking University campus (PKU) in Beijing, indicating offline FIGAERO-CIMS analysis can be more quantitative with appropriate calibrations (shown in Fig. S12 (Zheng et al., 2021))."

[Figure]

**Figure R10 (revised Figure S12).** Comparison between CHOX mass concentrations from FIGAERO-CIMS, organic aerosols (OA), and secondary organic aerosols (SOA) derived from ToF-ACSM at the Peking University Campus (PKU) site. Calibrations for FIGAERO-CIMS were conducted for a series of chemical compounds with both the permeation tube and micro-syringes. The details of the site, comparison setting up, calibrations, and calculations can be found in Zheng et al. (2021). In total, CHOX accounts for about 32–60% of SOA measured by the TOF-ACSM in their study.

We also refer the reader to the response to reviewer 1´s comments (Comment 10) on this topic.

The authors find that (lines 540-542) "The variability in Tmax induced by varying PM loadings is within 5°C for 29% of compounds and within 15°C for 54% of all compounds for Quartz filters, and 35% and 57% of compounds, respectively, for Teflon samples." They also summarize (in the abstract) that "we find that Tmax can be determined with high repeatability for one filter type". Taken together, this seems to imply that e.g. a 10°C difference in Tmax (due to filter loading) is acceptable. Is that the case? What volatility difference is associated with a 10°C difference in Tmax? Is that uncertainty / variability acceptable for volatility analysis?

Reply: We thank the reviewer for pointing out the implications of our statements regarding volatility determination using $T_{max}$, which were not fully intentional from our side. A ~5°C difference in $T_{max}$ from the duplicate tests would translate into less than one order of magnitude in the saturation concentration (C*) estimation (69% – 30% changes to the original C*) from different empirical equations in previous studies (Ylisirniö et al., 2021; Bannan et al., 2018; Lopez-Hilfiker et al., 2014; Stark et al., 2017; Nah et al., 2019; Ye et al., 2019)). However, more recent studies (Huang et al., 2018; Ylisirniö et al., 2021; Wu et al., 2021) present the challenges of determining the volatility of SOA particles through thermogram analysis and $T_{max}$, indicating that such analyses may (for complex mixtures with not fully defined phase state such as SOA particles) merely be qualitative. It is not within the scope of this study to assess the relationship between $T_{max}$ and volatility as such. Our goal here is to show the reproducibility of thermograms in general for individual filter samples and especially for the thermograms resulting from non-uniform ramping procedures. In the duplicate tests from the same sample, the majority of compounds (52%–70%) have $T_{max}$ difference within 5 °C (Line 597, Fig.9b).

We added additional information in the revised manuscript (Line 642 – Line 646):

"The variability in $T_{max}$ induced by varying PM loadings is within 5°C for 29% of compounds, and within 15°C for 54% of all compounds for Quartz filters, and 35% and 57% of compounds, respectively, for Teflon samples. The higher $T_{max}$ variation for different OA loading samples compared to the duplicate samples (±5.7°C, Fig.9 b) is likely caused by other factors, such as particle viscosity, the particles on the filter, and/or mass loadings on the filter (Huang et al., 2018; Ylisirniö et al., 2021; Wu et al., 2021; Graham et al., 2022)."

We also changed the expression in the abstract (Line 36 – Line 37):

"While we find that $T_{max}$ can be determined with high repeatability (±5.7°C) from the duplicate tests for one filter type,"

Editorial comments:

- Line 62: The FIGAERO-CIMS data from HOMEChem was recently published in AS&T: https://doi.org/10.1080/02786826.2022.2133593.

Reply: Thanks for the information. The reference has been added to the revised manuscript and Line 62 has been revised as follows:

"Having the advantage of combining molecular composition and volatility information, the FIGAERO-CIMS has been widely used for measuring OA compounds in many different environments including e.g. forests (Lopez-Hilfiker et al., 2016; Lee et al., 2016; Lee et al., 2018; Mohr et al., 2019), rural and urban areas (Le Breton et al., 2019; Huang et al., 2019; Cai et al., 2022), indoor air (Farmer et al., 2019), and cooking emissions (Masoud et al., 2022)."

- Line 312: I suggest replacing "right" with a different word (and maybe reorganizing the sentence); e.g. "This shows the importance of correctly assessing instrument background…"

Reply: Line 350 has been revised and rephrased as suggested.

"This shows the importance of correctly assessing the instrument background, especially for compounds with low signal."

- Fig 4d) – should the horizontal axis also be "Quartz"?

Reply: The reviewer is correct, the label of Fig. 4 b) has been corrected as suggested:

[Figure]

**Figure R11 (revised Figure 4)**. Comparison of the integrated signals from duplicate tests of the same 24-h sample for (a) Teflon and (b) Quartz fiber filters. The relative error (*Is* ratio of standard deviation/average) value of the 3 duplicate tests as a function of *Is* for (d) Teflon and (d) Quartz filters. In (c) and (d), CHOX compounds are shown as dots, inorganics as well as contaminants as squares colored by the m/z. The black cycles in (c) and (d) represent median values of signal intensity bins (with log Is intervals of 0.3 for the Is range of 0 to 2) and error bars represent the 25th and 75th percentile of binned values of Std(*Is*)/Avg(*Is*) for CHOX.

- Fig 6b) – should horizontal axis be "Q-punch"?

Reply: The reviewer is right. The label of Fig. 6 (b) has been corrected as suggested.

[revised manuscript text omitted]